# Regulatory T cells use heparanase to access IL-2 bound to extracellular matrix in inflamed tissue

Hunter A. Martinez [1], Ievgen Koliesnik [1], Gernot Kaber [1], Jacqueline K. Reid[2,3,4], Nadine Nagy [1], Graham Barlow [1], Ben A. Falk[5], Carlos O. Medina[1], Aviv Hargil[1], Svenja Zihsler[1], Israel Vlodavsky [6], Jin-Ping Li[7], Magdiel Pérez-Cruz[1], Sai-Wen Tang[1], Everett H. Meyer[1], Lucile E. Wrenshall[8], James D. Lord[9], K. Christopher Garcia [10], Theo D. Palmer [11], Lawrence Steinman [12], Gerald T. Nepom[13], Thomas N. Wight[5], Paul L. Bollyky [1,15] & Hedwich F. Kuipers [2,3,4,14,15] ✉

Although FOXP3+ regulatory T cells (Treg) depend on IL-2 produced by other cells for their survival and function, the levels of IL-2 in inflamed tissue are low, making it unclear how Treg access this critical resource. Here, we show that Treg use heparanase (HPSE) to access IL-2 sequestered by heparan sulfate (HS) within the extracellular matrix (ECM) of inflamed central nervous system tissue. HPSE expression distinguishes human and murine Treg from conventional T cells and is regulated by the availability of IL-2. HPSE$^{-/-}$ Treg have impaired stability and function in vivo, including in the experimental autoimmune encephalomyelitis (EAE) mouse model of multiple sclerosis. Conversely, endowing monoclonal antibody-directed chimeric antigen receptor (mAb-CAR) Treg with HPSE enhances their ability to access HS-sequestered IL-2 and their ability to suppress neuroinflammation in vivo. Together, these data identify a role for HPSE and the ECM in immune tolerance, providing new avenues for improving Treg-based therapy of autoimmunity.

In healthy individuals, immune tolerance is maintained by populations of regulatory T cells, including FOXP3+ regulatory T cells (Treg)[1]. However, in patients with autoimmune diseases such as multiple sclerosis (MS), Treg fail to control the autoreactive immune response central to their pathology. Although the frequency of circulating Treg is not different in MS patients compared to healthy controls, patient-derived Treg exhibit loss of in vitro function, as well as reduced FOXP3 expression levels[2–5]. Moreover, numbers of Treg within MS lesions have been reported to be low or non-existing, despite the presence of CD4+ T cell infiltrates[6]. This suggests that factors that regulate local

[1]Department of Medicine, Stanford University School of Medicine, Stanford, CA, USA. [2]Department of Clinical Neurosciences, Cumming School of Medicine, University of Calgary, Calgary, AB, Canada. [3]Hotchkiss Brain Institute, Cumming School of Medicine, University of Calgary, Calgary, Canada. [4]Snyder Institute for Chronic Diseases, Cumming School of Medicine, University of Calgary, Calgary, Canada. [5]Matrix Biology Program, Benaroya Research Institute, Seattle, WA, USA. [6]Technion Integrated Cancer Center, Technion, Haifa, Israel. [7]Department of Medical Biochemistry and Microbiology, Uppsala University, Uppsala, Sweden. [8]Department of Neuroscience, Cell Biology, and Physiology, Boonshoft School of Medicine, Wright State University, Dayton, OH, USA. [9]Translational Research Program, Benaroya Research Institute, Seattle, WA, USA. [10]Department of Molecular and Cellular Physiology, Stanford University School of Medicine, Stanford, CA, USA. [11]Department of Neurosurgery, Stanford University School of Medicine, Stanford, CA, USA. [12]Department of Neurology and Neurological Sciences, Stanford University School of Medicine, Stanford, CA, USA. [13]Immune Tolerance Network, Benaroya Research Institute, Seattle, WA, USA. [14]Department of Cell Biology and Anatomy, Cumming School of Medicine, University of Calgary, Calgary, Canada. [15]These authors contributed equally: Paul L. Bollyky, Hedwich F. Kuipers. ✉e-mail: hedwich.kuipers@ucalgary.ca

Treg homeostasis and function may contribute to the loss of tolerance central to autoimmune conditions.

One factor that governs Treg homeostasis is the cytokine interleukin 2 (IL-2). Although IL-2 is indispensable for Treg, they cannot produce it themselves[7–9]. IL-2 levels in blood, cerebrospinal fluid, and inflamed tissue are typically far lower than the concentrations required to support Treg survival and function in vitro[10–15]. Moreover, circulating IL-2 has a very short half-life (6–20 min)[16]. Given this poor availability of IL-2, it is unclear how Treg access this critical resource in vivo at sites of autoimmunity.

Several cytokines are sequestered within the extracellular matrix (ECM) by binding to HS-containing proteoglycans (HSPG), including murine and human IL-2[17–21]. In addition, binding to HS has been shown to potentiate the impact of IL-2 on both antigen presentation[21] and the pro-apoptotic effects of high-dose IL-2[20]. Unlike Treg, conventional T cells (Tconv) do not require exogenous IL-2 in vivo[22]. Therefore, the functional relevance of tissue-bound IL-2 for the regulation of T cell responses has been unclear.

Here, we test the hypothesis that FOXP3+ Treg use heparanase (HPSE) to access HS-bound IL-2 from the ECM within inflamed tissues and that this supports Treg homeostasis and function. We performed these studies in the experimental autoimmune encephalomyelitis (EAE) mouse model of multiple sclerosis, in which IL-2 and FOXP3+ Treg are known to modulate disease severity[23,24]. Our data reveal a previously unsuspected role for the tissue microenvironment in the regulation of Treg homeostasis and immune tolerance.

## Results

### IL-2 is bound to HS in EAE lesions

To explore the role of HS in autoimmune neuroinflammation, we asked whether IL-2 and HS are present at sites of inflammation in C57Bl/6/MOG$_{35-55}$ -induced EAE. We first characterized the distribution of both IL-2 and HS in spinal cord tissue from naïve (healthy) and EAE animals. We observed that naïve spinal cord tissue has scattered IL-2 staining. This staining is not associated with HS staining, which is found mostly in vasculature and meninges (Fig. 1a). In contrast, spinal cord tissue from EAE mice shows increased IL-2 staining throughout areas of parenchymal inflammation, as well as increased HS deposition (Fig. 1a, b). Both IL-2 and HS staining were found on cellular structures with astroglial morphology (Supplementary Fig. 1; Supplementary Movie 1), and in the form of diffuse staining in between cellular structures, suggesting association with peri- and extracellular matrix (Supplementary Fig. 1 Supplementary Movie 1). HS staining was also seen associated with some CD45 cells, both in perivascular spaces, as well as infiltrated into the CNS parenchyma. However, these cells generally did not stain positive for IL-2 (Supplementary Fig. 1). Staining for glial fibrillary acidic protein (GFAP), which stains the cytosekeleton of astrocytes, revealed that indeed, much of the cellular pattern of IL-2 and HS staining overlapped, or was directly adjacent to, GFAP+ astrocytic cytoskeletal structures, indicating that IL-2/HS deposition in EAE lesions is associated with reactive astrocytes (Supplementary Figs. 2, 3).

We then quantified the abundance and colocalization of IL-2 and HS throughout the course of EAE, using Imaris software. We observed that IL-2 staining was significantly increased in areas with CD45+ leukocyte infiltration (lesions), at the late acute stage of disease (Fig. 1c). HS was markedly increased in these areas as well, but also in areas adjacent to lesions (perilesion) and the glia limitans underlying the meninges, at the chronic stage of disease (Fig. 1c). This gradual increase correlated with the colocalization of IL-2 with HS, which was significantly increased in lesion, as well as perilesional areas at late acute and chronic stages (the percentage of IL-2 that is colocalized with HS ranging from 60 to 80%; Fig. 1c). These analyses show that the abundance of HS and IL-2 increase over the disease course of EAE and that these colocalize extensively.

To determine whether the deposition of IL-2 in inflammatory areas depends on its interaction with HS, we enzymatically removed HS structures using heparinase of bacterial origin and subsequently assessed the presence of both IL-2 and HS in the treated tissue. We found that heparinase treatment not only removes HS from the tissue, but also reduces IL-2 immunostaining where it overlaps with HS (Fig. 1d), as well as colocalization of IL-2 with HS (Fig. 1e), suggesting that IL-2 binds to HS in EAE lesions. Together, these data indicate that IL-2 is present and bound to HS at sites of autoimmune neuroinflammation.

### Binding to HS fragments increases IL-2 potency

We next asked whether binding to HS impacts the function of IL-2. We found that pre-incubation with soluble HS enhances the bioactivity of IL-2 as measured by proliferation of CTLL-2 cells, which depend on IL-2 for growth and viability (Fig. 2a). To specifically interrogate the role of the HS-binding capacity of IL-2 in this potentiating effect, we used a mutant of IL-2, T51P-IL-2, which contains a single point mutation that significantly decreases its ability to bind HS but leaves its soluble bioactivity intact[19]. Soluble IL-2 and soluble T51P-IL-2 have comparable effects on CTLL-2 proliferation (Supplementary Fig. 4a). However, pre-incubation of IL-2 with soluble HS enhances its biological activity 4- to 100-fold, with only modest effects on T51P-IL-2 (Supplementary Fig. 4b). These data demonstrate that binding to soluble HS amplifies the bioactive potency of IL-2.

To better assess the binding of IL-2 to HS and the biological implications of this binding, we coated magnetic beads with heparin, which is chemically similar to HS with differences in disaccharide composition and the level of sulfation[25], using click chemistry (Supplementary Fig. 5). We generated heparin coated beads as opposed to HS coated beads, because thiolated heparin is commercially available. We then loaded these beads with increasing concentrations of IL-2 and T51P-IL-2 and washed off unbound cytokine (Supplementary Fig. 5). Using CTLL-2 cell proliferation as a read-out (Fig. 2b), we compared the efficacy of IL-2- and T51P-IL-2-loaded beads to that of soluble IL-2 and T51P-IL-2. We found that heparin-coated beads loaded with IL-2 deliver biologically active cytokine in a dose-dependent fashion, whereas the effective dose delivered by T51P-IL-2-loaded beads is much lower (Fig. 2c and Supplementary Fig. 4c).

### Treg utilize HS-bound IL-2 for homeostasis

Using FOXP3.eGFP reporter mice, we next evaluated the impact of binding to HS on the ability of IL-2 to support in vitro homeostasis of CD4+/FOXP3+ Treg. We first observed that pre-incubation of IL-2 with HS enhances its support of the induction and survival of in vitro induced CD4+/FOXP3+ Treg (Fig. 2d), but not the survival of CD4+/FOXP3- Tconv in the same cultures (Fig. 2e). Next, we used our heparin-coated beads to compare the usage of HS-bound IL-2 between freshly isolated, enriched, CD4+/FOXP3+ Treg and CD4+/FOXP3- Tconv in the same cultures (Fig. 2f). We found that beads incubated with IL-2, but not T51P-IL-2, support in vitro survival of FOXP3+ Treg in a dose dependent manner but do not induce generation of new FOXP3+ cells (Fig. 2f, Supplementary Fig. 6a, b). In contrast, the viability of FOXP3- Tconv was not affected by either IL-2 or T51P-IL-2-loaded heparin beads, regardless of whether they expressed the IL-2 receptor CD25+ or not (Fig. 2g, Supplementary Fig. 6c).

To test whether Treg can remove IL-2 from heparin-coated beads, we co-incubated Treg with heparin-coated beads that were pre-incubated with either IL-2 or T51P-IL-2. We then separated these beads from the Treg and subsequently co-incubated them with CTLL-2 cells. By assessing the proliferative response induced by these beads we inferred how much IL-2 remained bound to these beads (Fig. 2h). As before, we found that IL-2-loaded beads, but not T51P-IL-2-loaded beads, induce proliferation of CTLL-2 cells (Fig. 2i). However, beads that were previously co-incubated with Treg

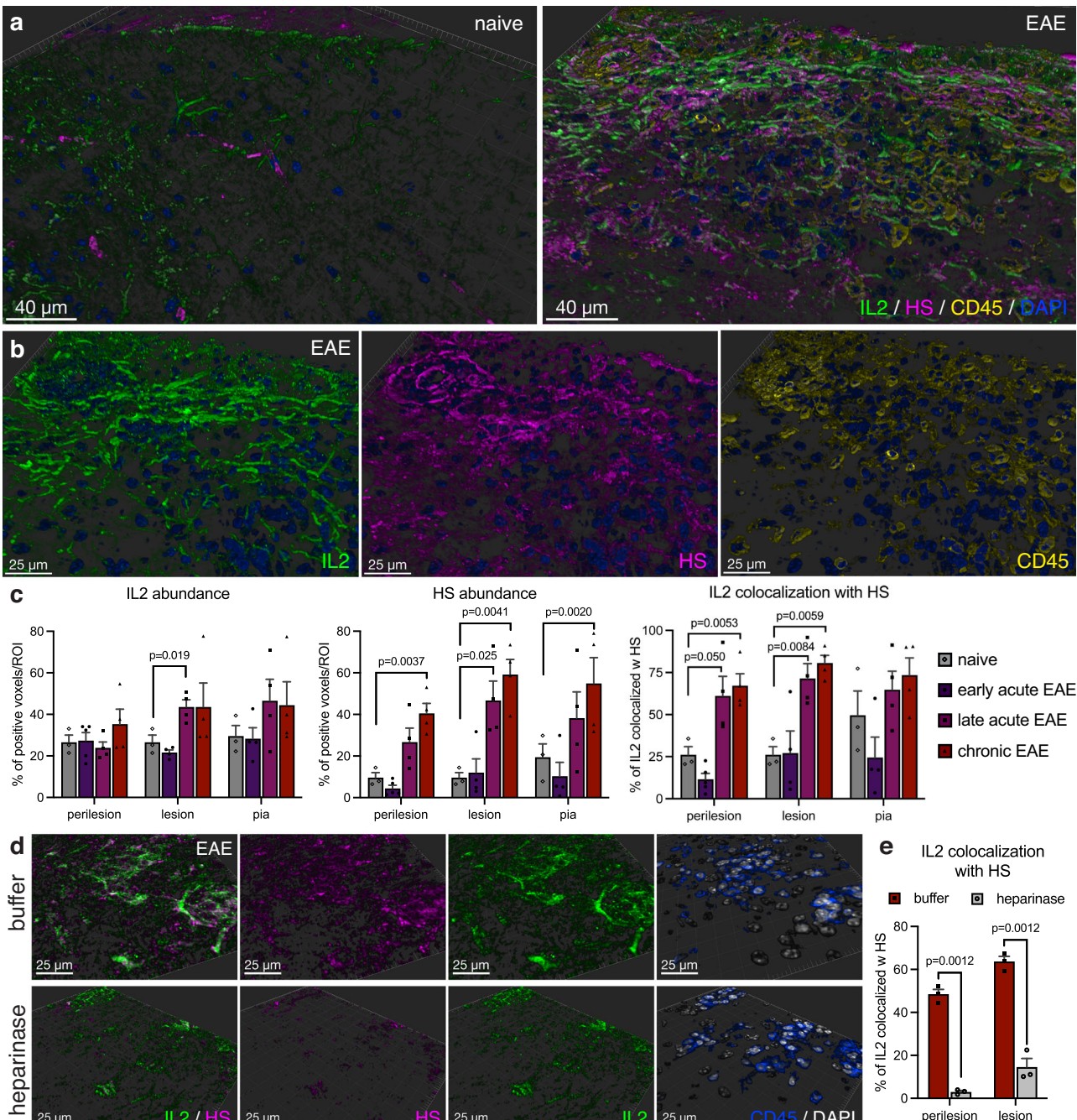

**Fig. 1 | IL-2 and HS colocalize at sites of autoimmune neuroinflammation.**
**a** Imaris surface rendering of immunofluorescent staining for IL-2 (green), HS (magenta) and CD45 (yellow) in naïve and EAE spinal cord tissue (29 days post immunization (dpi)). DAPI nuclear counterstain shown in blue. Representative of 3 separate EAE experiments. **b** Detail of IL-2 (green), HS (magenta) and CD45 (yellow) immunoreactivity in an EAE lesion. **c** Imaris quantification of abundance of IL-2 and HS immunoreactivity, and fraction of IL-2 colocalized with HS in naïve and EAE spinal cord tissue at different time points (early acute: 14 dpi, late acute: 29 dpi, chronic 40 dpi). Imaris colocalization analysis was done in areas surrounding CD45 cell infiltration (perilesion), areas with CD45 cell infiltration (lesion) and in the glia limitans underlying the meninges (pia). Shown are mean + SEM, $n = 3$ (naïve), 5

(early acute), 4 (late acute and chronic) separate areas analyzed. *P* values determined by unpaired two-tailed Welch's *t* test with Benjamini, Krieger and Yekutieli correction for multiple comparisons. **d** Immunofluorescent staining for IL-2 (green) and HS (magenta) of a cerebellar EAE lesion that was treated with heparinase from *Flavobacterium heparinum* before immunofluorescent staining, or buffer treatment of a serial section as a control. CD45 (blue) staining is shown to depict areas with immune cell infiltration. Representative of 2 separate experiments. **e** Imaris quantification of IL-2 colocalized with HS before (red) and after heparinase (gray) treatment. Shown are mean + SEM, $n = 3$ separate areas analyzed. *P* values determined by two-way ANOVA with Sidak's multiple-comparison correction.

promoted far less proliferation, indicating that Treg can remove IL-2 bound to heparin. Together, these data indicate that Treg, but not Tconv, use exogenous, HS-bound IL-2 to support their homeostasis. Moreover, these data suggest that Treg actively remove IL-2 that is bound to sequestered HS.

## Treg express higher levels of HPSE than Tconv

The data above suggest that Treg may actively engage HS-containing structures to access bound IL-2. HPSE is the only known enzyme that specifically binds and cleaves HS chains[26]. It was previously shown that Tconv express HPSE and that it is bioactive[27,28]. However, a role for

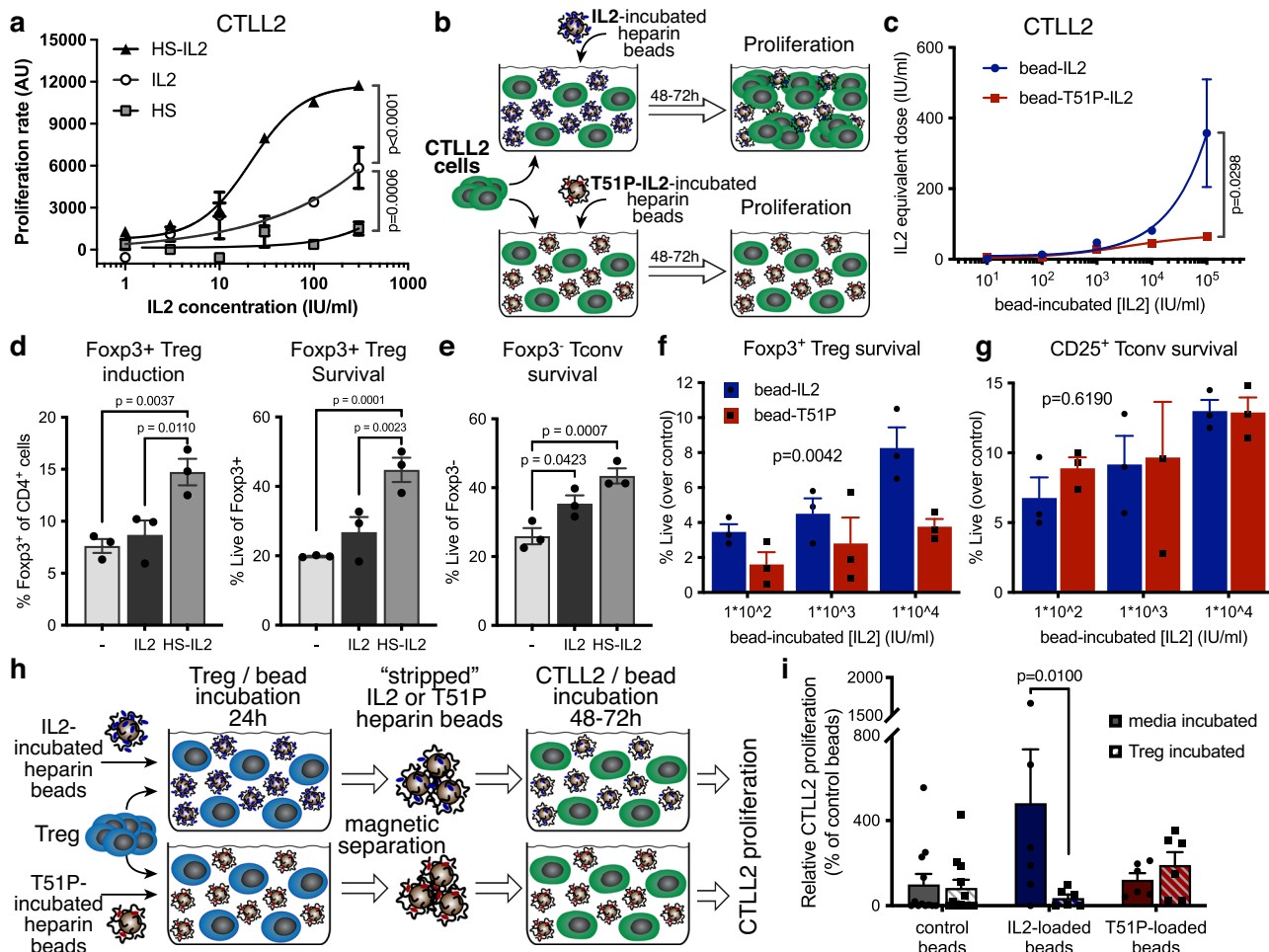

**Fig. 2 | HS-bound IL-2 supports Treg homeostasis. a** CTLL-2 proliferation, measured by resazurin reduction (arbitrary fluorescence units (AU)), in response to human recombinant IL-2 alone or pre-incubated with HS at a molecular ratio of 5:1. Equivalent doses of HS, not pre-incubated with IL-2, were used as a control. *P* values determined using two-way ANOVA. Shown is a representative of 4 independent experiments (mean +/− SEM of duplicate wells). **b** Schematic overview of CTLL-2 proliferation induced by heparin-coated beads pre-incubated with IL-2 or T51P-IL-2. **c** CTLL-2 proliferation in response to heparin-coated beads pre-incubated with IL-2 or T51P-IL-2. The concentration of IL-2 or T51P-IL-2 at which the beads were pre-incubated is depicted on the x-axis. Proliferation rate is shown as equivalent dose of soluble IL-2 or T51P-IL-2 at which a similar proliferative response is elicited as by the respective pre-incubated beads. Shown is a representative of 3 independent experiments (mean +/− SEM of triplicate wells). *P* value determined by two-way ANOVA. **d** Frequency and viability of FOXP3+ Treg induced from CD4+ T cells stimulated with anti-CD3/anti-CD28 in the presence of 50 ng/ml TGF-β and low-dose human recombinant IL-2 (20 IU/ml) alone, or IL-2 pre-incubated with HS (as described in Fig. 2a). **e** Viability of FOXP3- Tconv in the cultures described in (**d**). Cell frequencies and viability were measured by flow cytometry 72 hr after start of induction. Shown is a representative of 5 independent experiments (mean + SEM of triplicate samples), *P* values determined by one-way ANOVA with Tukey's multiple comparison correction. **f, g** Viability of FOXP3+ Treg (**f**) and CD25+/FOXP3- Tconv (**g**) among CD25+-enriched CD4+ T cells cultured in the presence of heparin-coated beads pre-incubated with IL-2 or T51P-IL-2 (T51P). Cells were analyzed by flow cytometry 24 h after start of culture. Percentage of viable cells among (**f**) FOXP3+ or (**g**) FOXP3- cells is depicted, corrected for baseline viability of cells cultured in media only (control: without IL-2). All panels show representatives of 4 independent experiments (mean + SEM of triplicate wells with cells pooled from 3-5 animals per experiment); *P* values depict variation due to the cytokine that the beads were incubated with (IL-2 vs. T51P-IL-2), determined by two-way ANOVA. **h** Schematic overview of the assay designed to assess the effectiveness of Treg-mediated stripping of IL-2 from heparin-coated beads. **i** CTLL-2 proliferation in response to IL-2 and T51P-IL-2-pre-incubated beads cultured with or without Treg. Proliferation rate is shown as percentage of proliferation in response to control beads, that were incubated in media in the first pre-incubation step. Shown is a representative of 3 independent experiments (mean + SEM of 6 wells per sample, cells pooled from 3-5 animals per experiment). *P* value determined by two-way ANOVA with Sidak's multiple comparison correction.

---

HPSE on Treg has not been studied. We therefore assessed the expression of HPSE in freshly isolated and sorted FOXP3+ Treg and FOXP- Tconv, both in resting cells and after in vitro activation.

Using real-time RT-PCR we found that upon activation murine FOXP3+ Treg express markedly higher levels of *HPSE* than FOXP3- Tconv (Fig. 3a). Western blot analysis revealed a similar increase at the protein level (Fig. 3b, c, Supplementary Fig. 7). Moreover, addition of exogenous IL-2 to the culture media reduced the activation-induced upregulation of *HPSE* mRNA expression in Treg, suggesting that *HPSE* expression is regulated by the availability of IL-2 (Supplementary Fig. 8a).

We similarly observed higher expression levels of *HPSE* in human in vitro activated FACS-sorted CD4+/CD25+/CD127- Treg, compared to sorted CD4+/CD25-/CD127+ Tconv (Supplementary Fig. 8b and Fig. 3d). Moreover, RNA sequencing of FACS-sorted Treg and Tconv present in blood and colon tissue samples from a registry of inflammatory gastrointestinal disease patients (Benaroya Research Institute Immune-Mediated Diseases Registry and Repository), including the autoimmune condition Crohn's Disease and Ulcerative Colitis[29], revealed that *HPSE* is differentially expressed in Treg compared to Tconv (Fig. 3e, f and Supplementary Fig 8d, e). Collectively, these data demonstrate that both Tconv and Treg

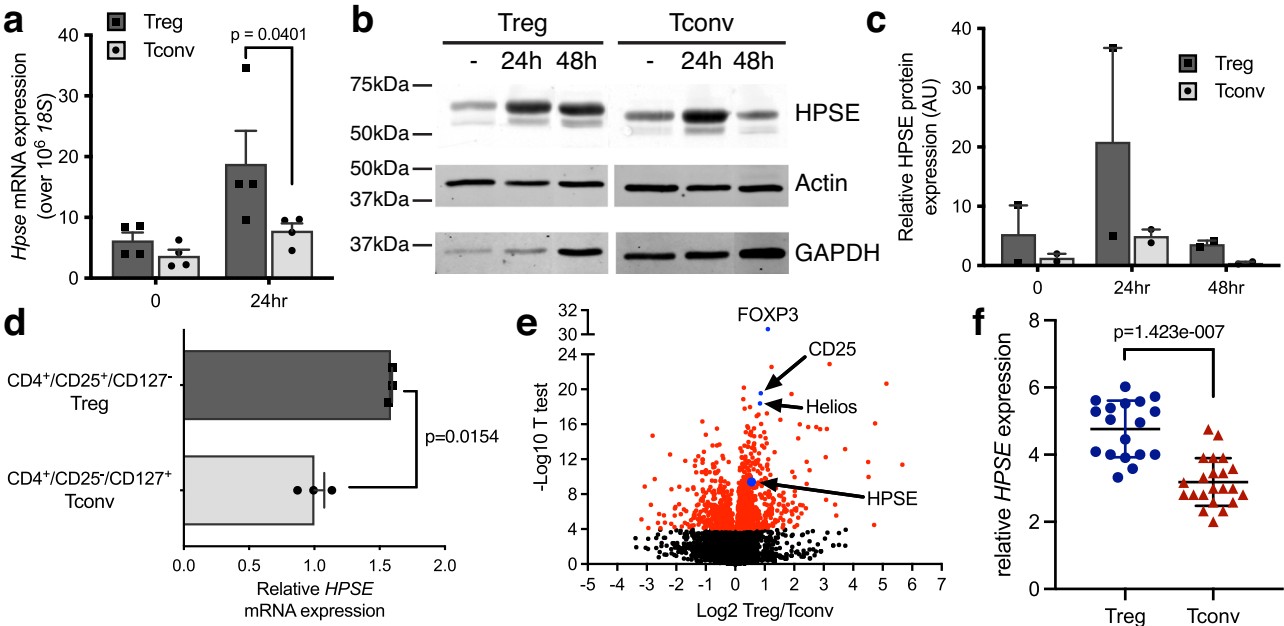

**Fig. 3 | Treg differentially express HPSE after activation. a** *HPSE* mRNA expression in FACS-sorted murine FOXP3+ Treg and FOXP3– Tconv after in vitro activation with aCD3/aCD28 antibody (24 h). Shown are mean relative expression + SEM, compared to resting Tconv and normalized by *18S* mRNA expression, n = 4 independent experiments performed with cells pooled from 5–10 animals per experiment, PCR ran with triplicate wells. Two-way ANOVA. **b** Western blot analysis and **c**, semi-quantitation of HPSE protein expression in murine Treg and Tconv after in vitro activation with aCD3/aCD28 antibody. Actin was used as a control to normalize quantitation. Shown are mean relative expression + SEM, compared to resting Tconv, n = 2 independent experiments, performed with cells pooled from 4 or 5 animals per experiment, single lanes ran per experiment. Two-way ANOVA with Dunnett's multiple comparison correction. **d** *HPSE* mRNA expression in FACS sorted and in vitro activated human CD4+/CD25+/CD127– Treg and CD4+/CD25–/CD127+ Tconv. Shown are mean relative *HPSE* expression + SEM of technical triplicates of a representative of 2 experiments, compared to Tconv and normalized by β-actin expression. *P* value determined by unpaired two-tailed *t*-test. **e** Volcano plot of statistical significance against fold change of genes differentially expressed between Treg and Teff isolated from human colon tissue. **f** *HPSE* mRNA expression in FACS sorted CD4+/CD25+/CD127– Treg and CD4+/CD25–/CD127+ Tconv isolated from human colon tissue. Shown are mean relative *HPSE* expression +/– SD, n = 18–22 sorted samples from individual subjects. *P* value determined by unpaired two-tailed *t*-test.

upregulate *HPSE* upon activation but that only in Treg do HPSE levels remain elevated.

## HPSE enhances the ability of Treg to access HS-bound IL-2

We then sought to determine the functional impact of HPSE on the utilization of HS-bound IL-2. To this end, we first stably overexpressed human HPSE in CTLL-2 cells, which already expresses low levels of murine HPSE (Fig. 4a, Supplementary Fig. 9). We then assessed their ability to respond to IL-2 bound to the heparin-coated beads described above. We found that HPSE overexpression results in an enhanced proliferative response to heparin-bound IL-2 compared to control CTLL-2 cells (Fig. 4b).

To interrogate the role of HPSE expression in Treg, we obtained HPSE-deficient (HPSE−/−) mice and crossed these to the FOXP3.eGFP reporter strain to generate HPSE−/− -FOXP3.eGFP mice. To facilitate comparisons of these strains, we backcrossed FOXP3.eGFP.HPSE−/− mice against FOXP3.eGFP.HPSE+/− mice for over 9 generations to generate matched FOXP3.eGFP.HPSE+/+ controls. We then assessed the ability of freshly isolated Treg from these strains to access heparin-bead-bound IL-2 in vitro. We found Treg isolated from HPSE−/− mice are unable to access bead-bound IL-2 as efficiently as Treg from WT mice (Fig. 4c). In contrast, Tconv are not impacted by the loss of HPSE (Fig. 4d and Supplementary Fig. 10a), probably because they can produce IL-2 endogenously[30].

Given the importance of IL-2/STAT5 signaling in Treg homeostasis[31,32], we next assessed the impact of HPSE expression on this signaling. We found that soluble IL-2 induces rapid STAT5 phosphorylation in both HPSE−/− and WT Treg (Supplementary Fig. 10b, c). However, when IL-2 is delivered bound to HS-coated plates, STAT5 phosphorylation is significantly reduced in HPSE−/− Treg, compared to

WT Treg (Fig. 4e), indicating that Treg depend on HPSE to access and respond to HS-bound, but not soluble IL-2. Moreover, we found that the frequency of pSTAT5+ cells among Treg is reduced in spleen tissue from HPSE−/− mice compared to WT mice (Supplementary Fig. 10d and Fig. 4f). This suggests that HPSE may promote tonic IL-2 responses in vivo as well.

## HPSE expression supports FOXP3+ Treg homeostasis in vivo

Next, we sought to define the role of HPSE in Treg homeostasis in vivo. We first examined the frequency of FOXP3+ Treg among CD4+ T cells in adult HPSE−/− and WT mice. This revealed that Treg frequencies are significantly lower in secondary lymphoid tissue (spleen and lymph nodes) of HPSE−/− mice, compared to WT mice (Fig. 4g, h and Supplementary Fig. 10e). We observed the same pattern in both HPSE−/−. FOXP3.eGFP on the C57Bl/6 background, as well as mice crossed with C57Bl/6 10BiT reporter mice from a different institution[33] (Supplementary Fig. 10f). Consistent with this, FOXP3+ Treg frequencies are reduced in secondary lymphoid tissue of HPSE−/− mice, but not in other tissues, including thymus, and the blood (Fig. 4h). Of note, the number and distribution of cytokine-producing CD4+ T helper cell subsets is not altered in lymphoid tissue of HPSE−/− mice (Supplementary Fig. 10g), indicating that these effects are specific to Treg. Moreover, whereas the frequency of FOXP3+ Treg increases with age in wild-type (WT) mice[34], in HPSE−/− mice, this aging-related increase in Treg frequency is significantly reduced (Fig. 4i). Together, these data suggest that Treg homeostasis is impaired in the absence of HPSE.

To test whether the effect of HPSE expression on the homeostasis of Treg is cell intrinsic, we performed two competitive transfer experiments. First, we generated mixed bone marrow chimeras using congenic (CD45.1+) WT and (CD45.2+) HPSE−/− donors and assessed the

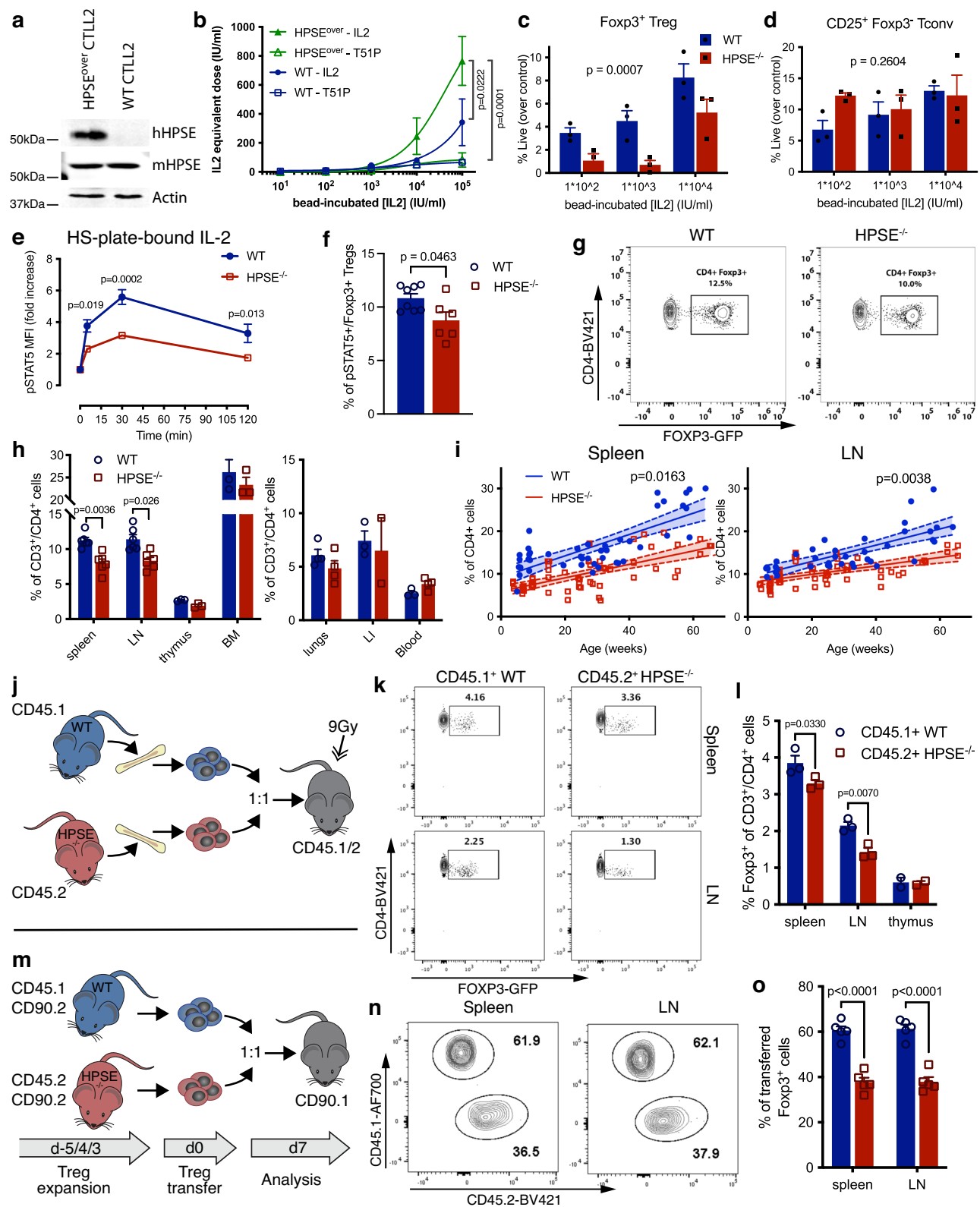

frequency of FOXP3+ Treg among donor-derived cells in lymphoid tissue of CD45.1/2 recipients (Fig. 4j). This revealed that Treg frequencies were significantly lower among CD4+ T cells derived from HPSE−/− donor cells than those derived from WT donor cells in recipient secondary lymphoid tissue, but not in the thymus (Fig. 4k, l).

Additionally, we FACS-sorted FOXP3+ Treg from CD45.1+ WT and CD45.2+ HPSE−/− donor mice, both on the CD90.2 background, and

adoptively transferred these cells into congenic CD90.1 recipient mice (Fig. 4m). Seven days post-transfer, the frequency of HPSE−/− donor-derived cells among transferred FOXP3+ Treg was significantly lower than that of WT donor-derived cells in both spleen and lymph node tissue (Fig. 4n, o and Supplementary Fig. 10h, i).

Together, these data indicate that the impairment seen in HPSE−/− Treg homeostasis is intrinsic to hematopoietic cells. Moreover, our

**Fig. 4 | HPSE expression supports FOXP3⁺ Treg homeostasis in vitro and in vivo.**
**a** Western blot analysis of human HPSE overexpression and mouse HPSE basal expression in CTLL-2 cells transfected with HPSE construct (HPSE$^{over}$), compared to untransfected CTLL-2 cells (WT). **b** Proliferation of WT CTLL-2 cells (WT, blue) and HPSE-overexpressing CTLL-2 cells (HPSE$^{over}$, green) in response to heparin-coated beads that were pre-incubated with IL-2 (closed symbols) or T51P-IL-2 (T51P, open symbols). The concentration of IL-2 or T51P-IL-2 at which the beads were pre-incubated is depicted on the x-axis. Proliferation rate is depicted as equivalent dose of soluble IL-2 or T51P-IL-2 at which a similar proliferative response is elicited as by the beads. Shown is a representative of 3 independent experiments (mean +/− SEM of triplicate samples). *P* values of variation due to the genotype of the cells are shown, determined with two-way ANOVA with Tukey's multiple comparison correction. **c**, **d** Viability of WT and HPSE$^{-/-}$ FOXP3⁺ Treg (**c**) and CD25⁺/FOXP3⁻ Tconv (**d**) among CD4⁺ T cells cultured in the presence of heparin-coated beads pre-incubated with IL-2. Viability was measured by flow cytometry 24 h after start of culture. Shown are representatives of 4 independent experiments (mean + SEM of triplicate samples); *P* values depict variation due to the genotype (WT vs. HPSE$^{-/-}$), determined by two-way ANOVA. **e**, Quantification of STAT5 phosphorylation by flow cytometry in WT and HPSE$^{-/-}$ FOXP3⁺ Treg after stimulation with IL-2 that was sequestered by plate-bound HS. Shown are mean +/− SEM of technical triplicates of a representative of 4 experiments. *P* values determined by two-way ANOVA with Sidak's multiple comparison correction. **f** Percentage of pSTAT5⁺ cells among CD4⁺/FOXP3⁺ Treg isolated from naïve WT and HPSE$^{-/-}$ spleen tissue. Shown are mean + SEM, *n* = 6-8 animals per group. *P* value determined by unpaired two-tailed *t*-test.

**g** Representative flow cytometry plots depicting FOXP3⁺ Treg frequencies among CD4⁺ T cells in spleen tissue of WT and HPSE$^{-/-}$ mice. **h** Quantification of FOXP3⁺ Treg frequencies among CD4⁺ T cells in lymphoid (left panel) and non-lymphoid (right panel) tissues of WT and HPSE$^{-/-}$ mice. BM, bone marrow; LI, large intestine. Show are mean + SEM, *n* = 3-6 animals per group. *P* values determined by unpaired two-tailed *t*-test with Holm-Sidak multiple comparison correction. **i** Linear regression analysis of Foxp3⁺ Treg frequencies among CD4⁺ T cells in the spleens (left panel) and inguinal lymph nodes (right panel) of WT and HPSE$^{-/-}$ mice during aging. *P* value shown comparing slopes, *n* = 44, 38 (WT spleen, LN), and 61, 60 (HPSE$^{-/-}$ spleen, LN) individual mice. **j** Schematic overview of competitive bone marrow transplantation of WT and HPSE$^{-/-}$ donors. **k** Representative flow cytometry plots depicting FOXP3⁺ Treg frequencies among transferred CD4⁺ T cells recovered from spleen and LN tissue of mice engrafted with bone marrow from WT and HPSE$^{-/-}$ mice. *n* = 1 representative mouse for each group. **l** Frequencies of FOXP3⁺ cells among WT and HPSE$^{-/-}$ bone marrow-derived CD4⁺ T cells in lymphoid tissues of irradiated recipient mice. *n* = 3 (spleen, LN) and 2 (thymus) individual mice. **m** Schematic overview of competitive Treg transfer of WT and HPSE$^{-/-}$ donor mice. **n** Representative flow cytometry plots depicting CD45.1⁺ and CD45.2⁺ cell frequencies among transferred Treg recovered from spleen and LN tissue of recipient mice. *n* = 1 representative mouse. **o** Frequencies of CD45.1⁺ WT and CD45.2⁺ HPSE$^{-/-}$ among Treg in lymphoid tissues of recipient mice. *n* = 5 individual mice per group. **l**, **o** Shown are mean + SEM, *P* values determined by two-way ANOVA with Sidak's multiple comparison correction.

transfer studies with purified Treg establish that these effects have specific relevance to these cells in this model system.

## HPSE expression enables FOXP3⁺ Treg to access tissue-bound IL-2

Considering our observation that overlapping IL-2 and HS staining in EAE lesions appears to be astrocyte-associated (Supplementary Fig. 2, 3), we next questioned whether Treg are capable of accessing IL-2 that is associated with astrocytes and whether HPSE is necessary for this. To test this, we pre-incubated confluent cultures of the astroglioma cell line U87-MG, which express HSPG on their cell surface[35], with IL-2. After washing off any unbound IL-2, we co-cultured these with freshly isolated CD4⁺ T cells from WT and HPSE$^{-/-}$ mice (Fig. 5a). We found that FOXP3⁺ Treg only survive in co-cultures with astrocytes when they have been pre-incubated with IL-2 and that this survival is significantly reduced in HPSE$^{-/-}$ Treg compared to WT Treg (Fig. 5b). In addition, heparinase treatment of the IL-2 pre-incubated reduced Treg survival, confirming the association of IL-2 with HS on the surface of these astrocytes (Supplementary Fig. 11a). By contrast, Tconv survival is supported by astrocytes regardless of whether they have been pre-incubated with IL-2, and this is not affected by HPSE expression (Fig. 5c and Supplementary Fig. 11b, c).

We also tested whether Treg are capable of utilizing IL-2 associated with CNS tissue. To this end, we pre-incubated homogenized spinal cord tissue from healthy mice with IL-2 and, after washing off unbound IL-2, co-cultured this with CD4⁺ T cells. Using these preparations, we observed that IL-2-pre-incubated spinal cord tissue supports the survival of WT Treg in an IL-2 dose-dependent manner and that this is significantly reduced in HPSE$^{-/-}$ Treg (Fig. 5d). Neither CD25⁺ nor CD25⁻ Tconv survival is enhanced by IL-2 pre-incubation of spinal cord tissue and does not depend on HPSE (Fig. 5e and Supplementary Fig. 11d). Consistent with this, HPSE-overexpressing CTLL-2 cells are also able to proliferate, in a dose-dependent manner, in response to spinal cord tissue pre-incubated with IL-2, but not in response to T51P-IL-2 pre-incubated spinal cord tissue (Supplementary Fig. 11e, f). However, control CTLL-2 do not display this proliferative response to tissue-associated IL-2 (Supplementary Fig. 11e, f), suggesting that perhaps, in contrast to the assays with heparin-beads or HS-incubated IL-2 described before, expression of HPSE is critical for CTLL-2 cells to access IL-2 that is bound to tissue.

Together, these data indicate that Treg are capable of accessing IL-2 that is associated with cells or tissue and that this is dependent on HPSE expression.

## HPSE expression supports FOXP3⁺ Treg function in vitro and in vivo

Finally, we questioned whether HPSE expression also affects Treg function in suppressing immune responses. We found that HPSE$^{-/-}$ Treg have impaired suppressive function relative to WT Treg in vitro (Fig. 5f). Furthermore, we observed that adoptively transferred HPSE$^{-/-}$ Treg are not able to suppress autoimmune neuroinflammation in vivo in the EAE mouse model of MS, whereas WT Treg are (Fig. 5g, h, Supplementary Fig. 12a, b and Supplementary Table 1). This reduced disease severity is mainly due to a higher recovery rate (more animals converting to a lower disease score) in mice that received WT Treg than in mice that either received HPSE$^{-/-}$ or no Treg (Supplementary Fig. 12b), consistent with the notion that Treg are involved in the recovery of EAE[36,37]. This treatment effect is correlated with a higher frequency of Treg and a lower frequency of overall CD4⁺ T cells in lymph nodes of mice that received WT Treg, compared to those that received HPSE$^{-/-}$ Treg (Supplementary Fig. 12c). In addition, donor WT FOXP3$^{GFP+}$ Treg can be found among T cells recovered from the spinal cord of recovering mice at the chronic stage of disease (d38), whereas donor HPSE$^{-/-}$ Treg are not present among these cells (Supplementary Fig. 12d). Together, these data indicate that HPSE facilitates the suppressive function of Treg in vitro and in vivo.

## HPSE over-expression supports the ability of FOXP3⁺ Treg to access HS-bound IL-2

Finally, to explore the translational potential of our findings, we assessed the impact of HPSE overexpression on the stability and function of chimeric antigen receptors (CAR) Treg stability and function, which show promise for the treatment of autoimmune diseases, such as MS[38,39]. For this, we used the flexible monoclonal antibody-directed CAR (mAbCAR) clone 1X9Q, which contains an anti-FITC scFv portion fused to CD28 and CD3ζ costimulatory domains[40], and fused the expression sequence of HPSE to this construct, as a proof-of-concept (Supplementary Fig. 13a, b). We first transfected mRNA produced from these constructs into CD25⁺ enriched Treg from WT C57Bl/6 mice, resulting in high transfection efficiency for both constructs (Supplementary Fig. 13c). When cultured with IL-2-loaded heparin-

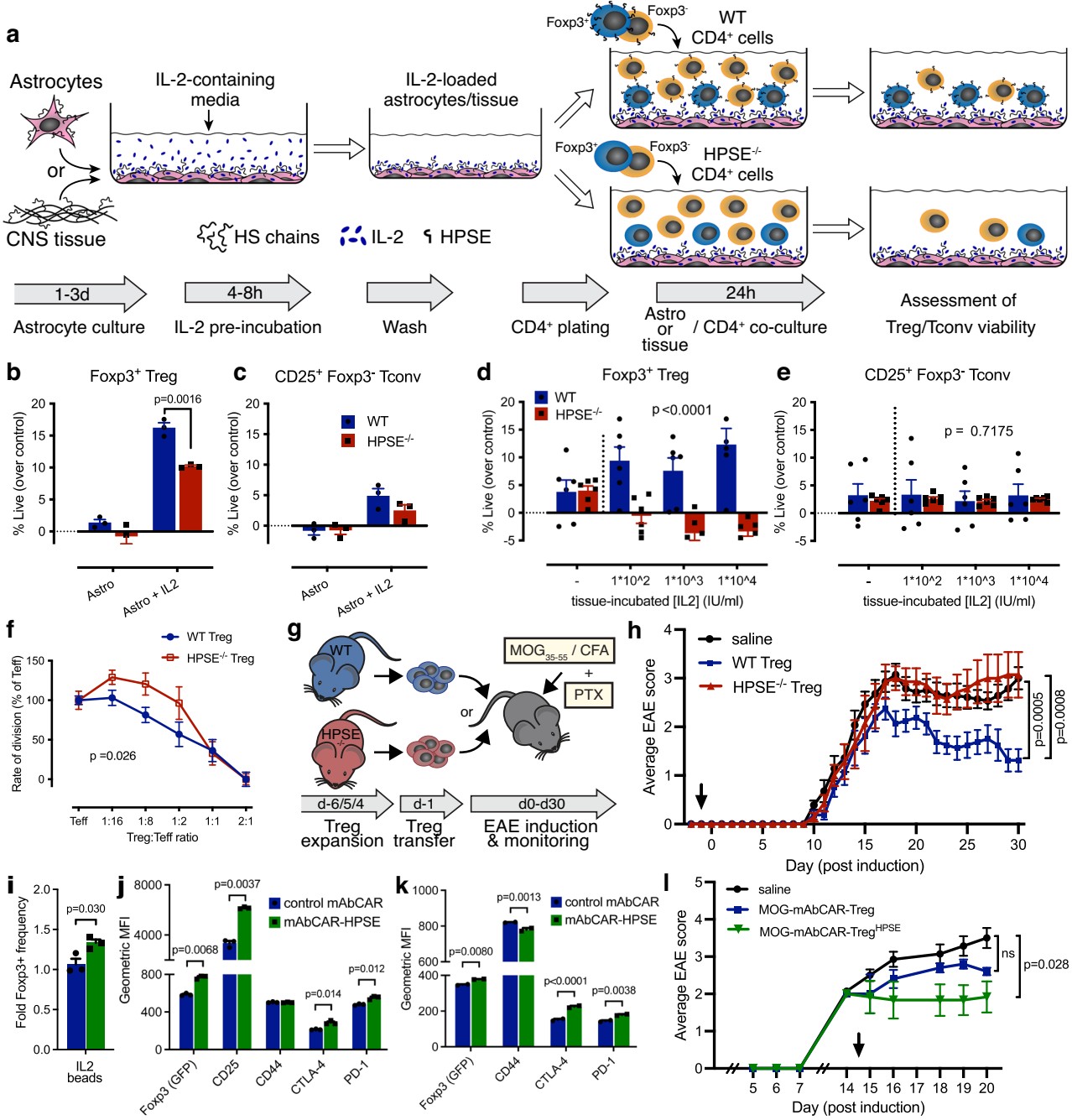

beads, we observed that persistence of FOXP3+ cells is marginally, but significantly, higher among these HPSE-overexpressing mAbCAR Treg than control mAbCAR Treg (Supplementary Fig. 13d and Fig. 5i). Moreover, cultured HPSE-overexpressing mAbCAR Treg have higher expression of not only FOXP3, but also the functional Treg markers CTLA-4 and PD-1 (Fig. 5j). Similarly, when CD45.1 FOXP3+ HPSE-overexpressing mAbCAR Treg were adoptively transferred into CD45.2 mice and recovered 3 days later, they exhibited slightly increased FOXP3 and co-inhibitory marker expression (Fig. 5k and Supplementary Fig. 13e, f). Finally, we targeted HPSE-overexpressing mAbCAR Treg towards myelin by incubating them with a FITC-conjugated monoclonal antibody against MOG (Supplementary Fig. 13b). We then transferred these MOG-specific mAbCAR Treg into EAE animals after onset of disease to test their ability to suppress established disease (Supplementary Fig. 13g). As expected in this challenging setting, control MOG-mAbCAR Treg do not significantly suppress disease. However, HPSE-overexpressing MOG-mAbCAR Treg significantly

reduce disease severity in these mice (Fig. 5l). Together, these data suggest that the expression of HPSE enhances the phenotypic stability and function of mAbCAR Treg.

## Discussion

In this study, we examined how FOXP3+ Treg obtain IL-2 at sites of autoimmune neuroinflammation. We report that IL-2 is sequestered by HS within the ECM and that FOXP3+ Treg use HPSE to access this essential resource to sustain their survival and function in vivo. In this way, ECM-bound IL-2, sequestered within inflamed tissue, may sustain regulatory responses in post-inflammatory settings. This mechanism may be particularly important in preventing and/or resolving auto-immunity, where Treg survival and function are critical for the main-tenance of immune homeostasis.

Our finding suggests that tissue interactions are vital to the effects of IL-2 on Treg. Binding to soluble HS fragments, which could be generated locally in vivo due to cleavage of ECM HS chains by HPSE,

**Fig. 5 | HPSE expression supports FOXP3⁺ Treg function in vitro and in vivo.**
**a** Schematic overview of experiments designed to assess the ability of mouse Treg to utilize IL-2 bound to U87-MG astrocytes or whole spinal cord (CNS) tissue in an HPSE-dependent manner. **b, c** Viability of WT and HPSE[-/-] murine (**b**) FOXP3⁺ Treg and (**c**) CD25⁺/FOXP3⁻ Tconv among CD4⁺ T cells co-cultured with human U87-MG cells that were pre-incubated with IL-2 or media alone as a control. **d, e** Viability of WT and HPSE[-/-] (**d**) FOXP3⁺ Treg and (**e**) CD25⁺/FOXP3⁻ Tconv among CD4⁺ T cells co-cultured with mouse spinal cord tissue that was pre-incubated with IL-2 or media alone as a control. **b–e** Percent of viable cells among FOXP3⁺ or FOXP3⁻ cells is depicted, corrected for baseline viability of cell cultured in media alone. All panels show mean + SEM ($n = 3$ (**b, c**) and 6 (**d, e**) separate wells) of a representative of 4 (**b, c**) or 3 (**d, e**) independent experiments using pooled cells from 3–5 animals per experiment. **b, c** $P$ values determined by unpaired two-tailed $t$-test with Holm-Sidak multiple comparison correction. **d, e** $P$ values depict variation due to genotype (WT vs. HPSE[-/-]), determined by two-way ANOVA. **f** In vitro suppression of WT effector (FOXP3⁻) T cells (Teff) by WT and HPSE[-/-] FOXP3⁺ Treg. The rate of division of Teff is plotted as percentage of unsuppressed Teff. Shown are mean +/− SEM ($n = 3$ separate wells) of a representative of 2 independent experiments, using pooled cells from 3–5 animals per experiment. $P$ value determined by two-way ANOVA between WT and HPSE[-/-] Treg. **g** Schematic overview of transfer of in vivo expanded, CD4⁺/CD25⁺ sorted, Treg into WT recipients and subsequent EAE

induction. **h** In vivo suppression of EAE by WT, and HPSE[-/-] FOXP3⁺ Treg that were transferred 1 day prior to induction of EAE (arrow). Control mice were injected with saline. Shown is average disease severity +/− SEM of all animals, $n = 14$ (control), 8 (WT Treg) and 7 (HPSE[-/-] Treg) animals, for one representative of 2 independent experiments. $P$ value determined by Friedman test with Dunn's correction for multiple comparisons. **i** Frequency and (**j**) expression levels of Treg functional markers of FOXP3⁺ cells among total CD4⁺ cells transfected with HPSE-mAbCAR after 72 h culture with IL-2-incubated heparin beads. Shown are mean + SEM ($n = 3$ separate wells) of the fold change of percentage FOXP3⁺ cells, relative to control mAbCAR cultured without IL-2, and of geometric MFI of each marker. **k** Expression levels of Treg functional markers on FOXP3⁺ cells recovered 3 days after adoptive transfer into CD45.2 recipients. Shown are mean + SEM ($n = 2$ independent experiments with 3–4 animals per group). **i** $P$ values determined by unpaired two-tailed $t$-test, and (**j, k**) two-way ANOVA with Sidak's multiple comparison correction. **l** In vivo suppression of EAE by control MOG-mAbCAR, and HPSE-overexpressing MOG-mAbCAR Treg that were transferred on day 14 after induction of EAE (arrow). Control mice were injected with saline. Shown is a representative of 2 independent experiments. Shown is average disease severity +/− SEM of all animals, $n = 7$ (control), 5 (MOG-mAbCAR) and 6 (HPSE-MOG-mAbCAR) animals per group. $P$ value determined by Friedman test with Dunn's correction for multiple comparisons.

---

enhances the biological effect of IL-2 on both FOXP3⁺ Treg and CTLL-2 cells, a T cell line exquisitely dependent on exogenous IL-2. Consistent with these effects, HS was reported to potentiate the effects of IL-2 on Tconv proliferation[20]. HS similarly potentiates the impact of other cytokines. HS-binding of IL-12 significantly reduces the $EC_{50}$ of this cytokine by acting as a co-receptor which enhances interactions between IL-12 and its receptor subunits[41]. IFN-γ interacts with HS and is thought to potentiate superfluous inflammation caused by excessive IFN-γ within tissues[42]. CXCL4 displays binding to endothelial HS proteoglycans to recruit leukocytes to the vasculature and increase vasculature permeability[43]. IL-21 bioavailability is regulated by binding to HS, dictating B cell germinal center differentiation and formation of antibody secreting cells[44]. Indeed, HS and other extracellular matrix components make up one third of tissue structures, demonstrating their relevance in tissue homeostasis[45]. In light of this potentiating effect of HS, and given the low levels of soluble IL-2 levels in blood, cerebrospinal fluid, and other fluids[10–14], HS-bound IL-2, liberated by HPSE, may provide a critical source of support for Treg homeostasis in vivo. Nevertheless, we do not claim that all IL-2 is associated with HS. Rather, we propose that by influencing the spatial and temporal properties of IL-2 in tissue, the ECM is a regulator of local immune homeostasis. These findings have clear implications for IL-2 as a therapeutic in autoimmunity and other inflammatory settings, and may inform future endeavors to design synthetic IL-2 or low-dose IL-2 treatment protocols, which have shown promise in clinical trials[46–48].

Competition for IL-2 is one of the mechanisms by which Treg suppress T-cell responses[31,33]. Therefore, the differential HPSE expression between Treg and Tconv may provide Treg with a competitive edge over effector T cells at sites of inflammation, where much of IL-2 is sequestered to HS in the ECM. Consistent with this, HPSE expression does not appear to be a benefit to Tconv, but rather play a critical role in Treg homeostasis and function.

The source of IL-2 in the CNS in the EAE model may be astrocytes given the colocalization of IL-2 with GFAP⁺ astrocytes in lesions. Consistent with this, astrocytes are reported to be able to produce both IL-2 and HS[35,49–51]. However, T cells, neurons and other cell types can also produce IL-2 and may contribute to the presence of IL-2 in these environments.

In this work, we have focused our efforts on targeted Treg studies, because of the complex phenotypes associated with systemic loss or inhibition of HPSE. However, HPSE is expressed by other cell types, including innate and adaptive immune cells[26,52,53] and may impact these cells in distinct ways. Future studies will interrogate these connections.

Likewise, distinct effects of HPSE on different cell types may underlie the conflicting data on HPSE inhibitors, some of which also have bioactivity independent of effects on HPSE[54]. Several such small molecules are promising therapies in cancer[55] while at least one was shown to worsen EAE[56]. It has also been shown that administration of exogenous HPSE reduces disease in EAE and increases the production of anti-inflammatory cytokines by T cells[57]. Overall, studies on the role of HPSE in MS and EAE are limited and challenging to interpret, given the multiple cell types and tissues involved. Further in-depth studies involving cell- and tissue-specific approaches are warranted to fully uncover the role of HPSE in MS.

Building on the insights described here, we provide proof-of-principle data that mAbCAR Treg engineered to express HPSE are better able to utilize HS-bound IL-2 and have enhanced phenotypic stability in vitro and in vivo. MOG-specific CAR Tregs have previously been shown to diminish disease severity in EAE upon intranasal transfer[58]. The results we present here demonstrate the promise of considering the cell-ECM interface in attempts to enhance the function of Treg-based therapies.

To conclude, we demonstrate that IL-2 is sequestered by HS at sites of autoimmune inflammation, that HS-bound IL-2 promotes FOXP3⁺ Treg survival and function, and that Treg require HPSE to access this essential resource (Supplementary Fig. 14a). Computational modeling of heparin (as a tetrasaccharide), using the ClusPro server[59,60], supports its binding potential for IL-2 and does not appear to interfere with the receptor binding to IL-2 (Supplementary Fig. 15). However, the heparin tetrasaccharide (~1 kDa molecular weight (MW)[61]) does not necessarily reflect the potential molecular weights of heparin and HS found within tissues (average MW ~30 kDa for HS, ~15 kDa for heparin[25]). Thus, while considering steric hindrance, multiple IL-2 molecules may be captured by heparin and more so by HS. With this knowledge, we hypothesis multiple effects are occurring. First, IL-2 is captured through electrostatic based interactions, with an estimated binding affinity being ~0.5 μM for HS[62]. However, avidity (synonymous with 'functional affinity') effects will allow reacquisition of IL-2 during on- and off-binding events. Concurrent with IL-2 binding to HS, heparanase (HPSE) on the cell surface captures the HS/IL-2 complex and either holds it near the surface or liberates it (Supplementary Fig 14b). Next, high affinity IL-2R alpha (CD25) captures the HS/IL-2 complex, which displays an affinity for IL-2 ~50× stronger at ~10 nM[63]. Therefore, the predominant consequence of IL-2 binding to HS, and HPSE expression on Treg, would be to locally enhance the concentration of IL-2 for acquisition by the IL-2R complex and

initiating IL-2 signaling. Together, these interactions resolve a critical gap in our understanding of Treg homeostasis and function in vivo. Therefore, these insights may provide new avenues for improving IL-2 and/or Treg treatment protocols.

## Methods

### Mice

To generate HPSE[-/-]-FOXP3.eGFP mice, harboring a eGFP-FOXP3 fusion reporter knock-in allele in addition to a targeted disruption of the HPSE gene, HPSE[-/-] mice, described before[64], were crossed with FOXP3.eGFP reporter mice[65], a kind gift of Dr. Alexander Rudensky. Both strains were maintained on the C57BL/6 J background and offspring was backcrossed and maintained as a homozygous line. In addition, resulting HPSE[-/-]-FOXP3.eGFP mice were crossed with 10BiT-FOXP3.eGFP mice, derived from crossing 10BiT mice containing a Thy1.1 (CD90.1) reporter under the control of the IL-10 promoter and FOXP3.eGFP reporter mice[66], a kind gift of Dr. Casey Weaver. Wild-type (CD45.2) (Strain #:000664) and congenic CD45.1 C57BL/6 J (Strain #:002014), C.129P2(B6)-Il2[tm1Hor]/J (IL2[-/-]) (Strain #:002229) and B6.129S7-Rag1[tm1Mom]/J (RAG1[-/-]) (Strain #:002216) mice were obtained from Jackson Laboratories (Bar Harbor, ME). Congenic CD90.1 C57BL/6 J were a kind gift of Dr. Robert Negrin. All mice were maintained in specific pathogen-free AAALAC-accredited animal facilities at the BRI and Stanford University and handled in accordance with institutional guidelines. Protocols used for experimentation were approved by the Stanford University Administrative Panel on Laboratory Animal Care. Male mice were used for all experiments, unless otherwise stated. Animals were maintained in a specific pathogen-free (SPF) facility. Experimental and control animals were housed in the same SPF facility, but in separate respective cages. Animals were in a 12-h light/dark cycle beginning at 7am (light) and 7 pm (dark). Ambient temperature was 20–22 °C with humidity 40–60%. Mice were euthanized using $CO_2$ asphyxiation dispensed from a fixed pressure regulator and inline restrictor controlling gas flow. $CO_2$ flow was maintained for at least 7 min. Death was verified following euthanasia by monitoring cessation of heartbeat and respiration, as well as a toe pinch reflex.

### Induction of EAE

EAE was induced in male C57BL/6 J mice (Jackson Laboratories) at 8–12 weeks of age by subcutaneous immunization with an emulsion containing 200 µg of myelin oligodendrocyte glycoprotein (35–55) ($MOG_{35-55}$) in saline and an equal volume of complete Freund's adjuvant (CFA) containing 400 ng of mycobacterium tuberculosis H37RA (Difco Laboratories, Detroit, MI). All mice were administered 400 ng of pertussis toxin (List Biological, Campbell, CA) intraperitoneally (i.p.) at 0 and 48 h post-immunization. Mice were weighed and monitored daily for clinical symptoms and scored as follows: 0, no clinical disease; 1, tail weakness; 2, hindlimb weakness; 3, complete hindlimb paralysis; 4, hindlimb paralysis and some forelimb weakness; 5, moribund or dead. At peak of disease (day 22), mice were sacrificed and brain tissue was harvested for immunohistochemical analysis.

### Histology

Mice were deeply anesthetized with ketamine/xylazine (100 mg/kg and 7 mg/kg, respectively), transcardially perfused (saline followed by 4% paraformaldehyde in PBS) and spleens, lymph nodes (inguinal, axillary and brachial) and brains were collected. Tissue was post-fixed overnight in 4% paraformaldehyde at 4 °C, then cryoprotected in 30% sucrose and stored at 4 °C until they were embedded in Optimal Cutting Temperature compound (Tissue-Tek) and frozen at −80 °C. Cryosections (20 µm) were stained using a free-floating immunohistochemistry protocol. Briefly, after blocking in 5% normal donkey serum (NDS) in Tris-buffered saline (TBS) with 0.3% Triton X-100 (TBS-T), sections were incubated overnight at 4 °C in TBS-T with 1% NDS and the following primary antibodies (Supplementary Table 2): mouse anti-HS (clone F58-10E4, Seikagaku; 1:50), chicken anti-IL-2 (Sigma GW22461F; 1:250) and rat anti-CD45 (clone 30-F11, Life Technologies MCD4500; 1:500). After 3 washes with TBS-T, sections were incubated with species-specific fluorescent secondary antibodies from donkey conjugated with Alexa Fluor 488, 594, or 647 (Jackson Laboratories; all 1:500) for 45 min at RT.

### Imaging and image processing

All images were acquired using an automated Zeiss CellDiscover7 LSM900 confocal microscope (Zeiss), using identical settings across all samples. z-stacks were acquired of regions of interest (inflammatory lesions or corresponding areas in control tissue) using an 20×/0.95 NA objective, at a voxel size of 0.097 × 0.097 × 0.63 µm. Channels were acquired sequentially to prevent spill-over. Images we pre-processed in Fiji (version 2.3.0; National Institutes of Health) using rolling ball background subtraction, with a rolling ball radius of 50.0 pixels. Pre-processed images were then analyzed using Imaris software (Bitplane). First, masks were made for each channel to enable surface rending and colocalization analysis, using identical thresholding and voxel minimum parameters for all samples. In addition, regions of interest (ROIs) were set across multiple z planes for areas with CD45 cell infiltration ("lesion"), areas adjacent to lesions ("peri-lesion") and the glia limitans underlying the meninges ("pia"). Then colocalization of IL-2 and HS fluorescence was calculated using the Imaris colocalization algorithm within these ROIs. Data is represented as % of positive IL-2 or HS voxels within the ROI, as well as % of IL-2 positive voxels that are colocalized with HS.

### Heparinase treatment of tissue

20 µm cryosections were incubated with 50 mU Heparinase III from *Flavobacterium heparinum* (Sigma-Aldrich) in 10 mM HEPES and 2 mM $CaCl_2$, pH7.0 at 37 C for 1 h. As a control, tissue was incubated at the same conditions in buffer without heparinase. After HS digestion, sections were washed once in PBS and stained as described above.

### CTLL-2 culture and resazurin proliferation assays

For maintenance, CTLL-2 cells were cultured in media (RPMI 1640 containing 2.05 mM L-Glutamine, 10% FBS and 1× penicillin/streptomycin; GE Healthcare, Pittsburgh, PA) supplemented with recombinant human IL-2, produced in HEK cells, at EC75, which was established measuring proliferation in a dose-response assay. For proliferation assays, CTLL-2 cells were first washed twice with media not containing IL-2 and incubated in media not containing IL-2 for at least 4 h at 37 °C. Cells were then washed again and plated at a density of $0.5 \times 10^5$ cells per well in a total of 100 µl media. After 24 to 48 h, proliferation was assessed by reduction of resazurin to resorufin[67]. Resazurin (Acros Organics) stock solutions were made in RPMI 1640 at 5 mg/ml (w/v), filter sterilized and stored at −20 °C until use. Resazurin was added at a final concentration of 1 mg/ml to the cell cultures for the last 8 to 16 h of cell culture. Fluorescence of the resorufin was then measured at 560 excitation/590 emission using a Spark multimode microplate reader (Tecan).

### Generation of IL-2 and T51P-IL-2

Human recombinant IL-2 and the IL-2 analog in which the threonine at position 51 was replaced by a proline (T51P-IL-2) were expressed using a baculovirus system in Hi5 insect cells and purified as previously described[68]. Full vector sequences are available upon request.

### HS preincubation of IL-2

Human recombinant IL-2 (either Proleukin (Novartis Pharma), or in-house generated IL-2) was co-incubated at a molecular ratio of 5:1 with HS (Sigma Aldrich) for 8 h at RT, in a minimal volume of PBS. For stock solutions, 1 mg/ml HS and $1 \times 10^6$ U/ml IL-2 were co-incubated. Aliquots were frozen at −80 °C and diluted to working concentration in media prior to experiments.

### Generation and IL-2 loading of heparin-coated magnetic beads

Thiolated heparin (BioTime Inc.) was coupled to 1 µm BcMag™ epoxy-activated magnetic beads (Bioclone Inc.) as per manufacturer's directions to yield heparin-coated beads. These beads were then incubated with IL-2 or T51P-IL-2 in cell culture media (RPMI 1640 containing 10% FCS and penicillin/streptomycin) at indicated concentrations. Beads were washed 3 times with cell culture media before co-culture with cells at a 1:5 cells:beads ratio.

### Isolation and culture of mouse CD4⁺ T cells and Treg

Total leukocytes were isolated from inguinal, axial, brachial and mesenteric lymph nodes and spleen cells from 8 to 12-week-old male mice. Tissue was homogenized through a 70 µm strainer and red blood cells were lysed in the splenocyte suspensions. CD4⁺ T cells were isolated from the pooled cell suspensions using an EasySep™ Mouse CD4⁺ T Cell Isolation Kit (Stemcell Technologies, Vancouver, Canada) following the manufacturer's instructions. Treg were isolated from total cell suspensions using an EasySep™ Mouse CD25 Regulatory T Cell Positive Selection Kit (Stemcell Technologies). Cells were cultured in T cell media (RPMI 1640 containing 2.05 mM L-Glutamine, 10% FBS, 1× penicillin/streptomycin, 50 µM β-mercaptoethanol, and 1 mM sodium pyruvate), supplemented with IL-2 when indicated. For real-time PCR analysis of HPSE expression, FOXP3^GFP+ Treg were sorted from CD4⁺ T cell preparations using a BD FACSAriaII cell sorter. Viability analysis of CD4⁺ T cells under the experimental conditioned described in the figure legends was performed by staining with the amine-reactive dyes Zombie Red or Zombie Aqua (Biolegend), according to manufacturer's instructions, and flow cytometry.

### Flow cytometry

Single cell suspensions were prepared from spleen, lymph nodes, thymus and femurs of mice. Lungs and large intestines were minced and digested with 100 IU/ml Collagenase I and 25 IU/ml DNase I (both from Roche). Red blood cells were lysed in red blood cell lysis buffer (Sigma-Aldrich). For phosphoflow staining, cells were fixed by resuspending in 50 µl pre-warmed of Cytofix Fixation Buffer (BD Biosciences) and incubated at 37 °C in a CO₂ incubator for 15 min. Cells were then washed in FACS buffer (3% FBS (w/v) and 2 mM EDTA in PBS) and cell surface antigens were stained with anti-CD3-BV785 (17A2), anti-CD4-BV421 (RM4-5) and anti-CD25-APC-Cy7 (PC61) in FACS buffer for 30 min on ice. After washing, cells were then resuspended in pre-chilled Phosflow Perm Buffer III (BD Biosciences) and incubated on ice for 30 min. The cells were washed two more times with FACS buffer prior to staining using an antibody against pSTAT5 (Y694, BD Biosciences), diluted in FACS stain buffer on ice for 60 min. After staining, cells were washed once, resuspended in FACS stain buffer and analyzed. For cell viability analysis, 1 × 10⁶ cells were stained with anti-CD3-BV785 (17A2), anti-CD4-BV421 (RM4-5), anti-CD25-APC-Cy7 (PC61; all from BioLegend) and ZombieRed Live/Dead dye (Biolegend). For phenotyping and cytokine staining 5 × 10⁶ cells were stimulated with 50 ng/ml PMA and 500 ng ionomycin (Sigma-Aldrich) in the presence of Brefeldin A (Biolegend) for 5 h. Cells were then stained with anti-CD3-PE-Cy7 (17A2), anti-CD4-BV785 (RM4-5), and anti-CD8-BV711 (53−6.7), fixed using IC buffer (Thermo Fisher Scientific), permeabilized with 0.5% saponin in PBS (w/v) and stained with anti-IFNγ-APC (XMG1.12), anti-IL-17-PE (TC11-18H10.1) and anti-TNFα-PE (MP6-XT22). Antibody details can be found in Supplementary Table 2. Flow cytometry was performed on an LSRII (Becton Dickinson) in the Stanford Shared FACS Facility and data analysis was done using FlowJo 9 or 10 software (Treestar).

### Cell sorting

For mouse T cell sorting, cell suspensions were made from spleen and lymph node tissue. For human T cell sorting, PBMC were prepared from whole blood using Percoll gradient (GE Healthcare). Mouse and human CD4⁺ cells were enriched using an EasySep Mouse or Human CD4 isolation kit (Stemcell Technologies). Mouse cells were then sorted based on FOXP3^GFP expression. Human cells were stained with anti-CD4-FITC (RPA-T4; Biolegend), anti-CD127-PerCpCy5.5 (A019D5) and anti CD25-PE-Cy7 (M-A251, both BD Pharmingen) prior to sorting. Cells were sorted using a FACS Aria III (BD Biosciences) in the Stanford Shared FACS Facility and lysed in Trizol (Thermo Fisher Scientific) for RNA expression analysis.

### In vitro activation of T cells

96 well flat bottom tissue culture plates were pre-coated with 2.5 µg/ml anti-CD3 Ab (clone 145-2C11; Biolegend, San Diego, CA). FACS-sorted mouse CD4⁺/FOXP3⁺ or CD4⁺/FOXP3⁻ cells, and human CD4⁺/CD127/CD25⁺ Treg or CD4⁺/CD127⁺/CD25⁻ Tconv (for RNA analysis), or bead-sorted mouse CD25⁺ or CD25⁻/CD4⁺ cells (for protein analysis) were plated at a density of 2.5 × 10⁵ per well in T cell media containing 0.5 µg/ml anti-CD28 (clone 37.51; Biolegend). No exogenous IL-2 was added unless otherwise noted.

### Real-time PCR

After in vitro activation, cells were harvested and RNA was collected using an RNeasy mini kit (Qiagen). cDNA was prepared from 350 ng total RNA reverse transcribed in a 40 ul reaction mix with random primers using the High-Capacity cDNA Reverse Transcription Kit (Applied Biosystems), according to manufacturer's instructions. 1.2 ul cDNA was amplified in 1XTaqman Fast Universal PCR Mix with 250 nM Taqman probe (all Applied Biosystems) in a 20 ul reaction using the Fast program for 50 cycles on an ABI7900HT thermocycler. All samples were run in duplicate and data were analyzed using the Comparative Ct Method with software from Applied Biosystems. Estimated copy numbers were generated from a standard curve created by using a selected reference cDNA template and Taqman probe. For human T cells, qPCR was performed using SYBR® Green PCR Master Mix (Applied Biosystems). Primer sequences are available upon request. Human HPSE forward primer is 5′-AAGACGGCTAAGATGCTGAAG-3′ and reverse primer is 5′-CGTCCATTCAAATAGTAGTGATGC-3′. Mouse HPSE forward primer is 5′-GAGAGAGCCAGTAATCAGGTAAAG-3′ and reverse primer is 5′-GGGCATAGAAGTCGTGATGAG-3′.

### Western blot

After in vitro activation, CD25⁺ or CD25⁻ CD4⁺ cells were harvested and taken up in RIPA buffer with HALT protease and phosphatase inhibitor mix (Thermo Fisher Scientific). Lysates were then incubated on ice for 30 min, spun down at 13,000 g for 30 min at 4 °C and supernatants were transferred to a new eppendorf tube. Protein content was measured using a BCA assay (Pierce), according to manufacturer's directions. 25 ug of total protein was loaded onto a 12% PAGE gel. Proteins were transferred to nitrocellulose membranes and immunoblotted with primary antibodies against murine HPSE (rabbit polyclonal antibody PAA711Mu04, Cloud-Clone Corp.), human HPSE (mouse monoclonal antibody clone HP3/17, ProSpec), human/murine β-actin (mouse monoclonal antibody CAB340Hu22, Cloud-Clone) and murine GAPDH (mouse monoclonal antibody 6C5, EMD Millipore) and subsequently DyLight-680 or DyLight800-conjugated secondary antibodies (Supplementary Table 2). Antibody binding was visualized and quantified using and Odyssey Imaging system and Image Studio software (Li-Cor).

### Analysis of RNA expression data of human donor derived T cells

RNA sequencing data from CD25⁺/CD127/CD4⁺ Treg, and CD161⁺ and CD161⁻ CD45RA⁻/CD4⁺ T cells FACS-sorted from blood and colon tissue was obtained from the Benaroya Research Institute Immune-Mediated Diseases Registry and Repository, from Dr. James D. Lord[29]. Approval for blood and colon tissue collection was obtained through the Benaroya Research Institute Institutional Review Board (No. 10090). Relative expression levels of genes in Treg vs. Teff (Log2 Treg

expression / Teff expression, and -Log10 *T* test) were plotted as volcano plot. In addition, *HPSE* expression levels in these cell populations were obtained for all subjects and compared in Treg vs. CD161- Teff.

### CTLL-2 overexpression of HPSE
The murine HPSE transgene was co-expressed with an eGFP-Zeocin marker from a polycistronic message driven by the minimal EF1a promoter utilizing a self-cleaving P2A peptide. Lentiviral vectors were packaged in HEK 293 T cells as previously described. Briefly, 10 μg of transfer plasmid was cotransfected with helper plasmids (10 μg of pCMV-Pol/Gag, 4 μg of pCMV-Rev, and 4 μg of pCMV-VSVG) into HEK 293 T cells at 80% confluence using Fugene 6 (Promega). Viral supernatant was harvested 48 h post transfection, concentrated by ultracentrifugation, and stored at −80 °C until use. Viral titers were determined by transduction of HT1080 cells and analyzed for GFP expression by FACS. CTLL-2 cells were transduced with viral supernatants at an MOI of 10. Protransduzin-A (Immundiagnostik AG) was added according to the manufacturer's instructions and transduction reactions were spinoculated at 1000 g for 90 min at 32 °C in sealed tubes. An equal amount of fresh medium was added and cells were incubated in the presence of transduction complex for an additional 4 h at 37 °C. The transduction medium was replaced with fresh medium and cells were cultured for 48 h before starting selection with 200 μg/ml Zeocin. After selection for 7 days >99 % of cells were GFP positive by flow cytometry. Full vector sequences are available upon request.

### In vitro induction of HS-bound IL-2 signaling
96 well flat bottom tissue culture plates were pre-coated with subsequently anti-HS antibody (25 μg/ml) in PBS, 2.5 ug/ml HS in water, 10% BSA in PBS and 1000 IU/ml human recombinant IL-2. Wells were washed 3 times with PBS in between each incubation step. CD4$^+$ sorted cells were rested on ice in T cell media for 30 min before starting incubation by plating them on the pre-coated plates and gently spinning down for 30 s. After stimulation, cells were processed for phosphoflow staining as described above.

### Bone marrow chimeras
Transfer of bone marrow cells from WT and HPSE$^{-/-}$ donor mice was performed as described previously[69]. In brief, recipient 6–8-week-old male CD45.1/CD45.2 F1 heterozygote recipient mice were lethally irradiated with 9 Gy, rested for 24 h and intravenously injected with $5 \times 10^6$ bone marrow cells from WT and HPSE$^{-/-}$ mice at a 1:1 ratio. Chimerism was assessed 2 months after bone marrow transfer.

### Expansion and adoptive transfer of Treg
Expansion of Treg was induced in WT and HPSE$^{-/-}$ -FOXP3.eGFP mice by treatment with IL-2/anti-IL-2 antibody complex treatment as published before (Boyman et al., 2006). In short, complexes were prepared by incubating recombinant mouse IL-2 (eBioscience, 14-8021) and anti-IL-2 antibody (clone JES6-1, eBioscience, 16-7022) at a 1:5 ratio (w/w) for 30 min at 37 °C. After co-incubation, complexes were diluted with PBS and mice were administered 1 μg IL-2 / 5 μg anti-IL-2 i.p. on 3 consecutive days. On day 5 after the first injection, spleens and lymph nodes were harvested and CD4$^+$/FOXP3$^{GFP+}$ Treg were FACS sorted. For competitive adoptive transfer experiments, cells were washed, resuspended in PBS and mixed at a 1:1 ratio. A total of $2 \times 10^6$ CD25$^+$ cells were injected i.v. into 8–10-week-old male CD90.1 C57BL/6 J mice. Distribution of donor Treg cells was analyzed 7 days after transfer. For treatment of EAE, IL-2/anti-IL-2 antibody complex-expanded Treg were isolated using an EasySep™ mouse CD25 regulatory T cell positive selection kit (StemCell Technologies) according to manufacturer's directions. Cells were washed and resuspended in PBS and $1 \times 10^6$ WT or HPSE$^{-/-}$ CD25$^+$ cells were transferred i.v. in 100 ul to 8–10-week-old male C57BL/6 J mice one day before inducing EAE in the recipient mice as described above. Control mice were given an injection of saline. Mice were monitored daily for clinical

symptoms as follows: 0, no clinical disease; 1, tail weakness; 2, hindlimb weakness; 3, complete hindlimb paralysis; 4, hindlimb paralysis and some forelimb weakness; 5, moribund or dead.

### Astrocyte−T cell cocultures
U87-MG astrocytoma cells were cultured in DMEM (GE Healthcare) containing 10% FBS and 1× penicillin/streptomycin. When cells were confluent, they were pre-incubated with 1*10$^3$ IU/ml rhIL-2 or media for 4–8 h. After this, cells were washed 3 times with T cell media and freshly isolated murine CD4$^+$ T cells were plated on top. Cells were harvested for flow cytometry 24 h after coculture.

### In vitro Treg suppression assay
FACS sorted CD4$^+$/FOXP3$^{GFP+}$ Treg were titrated into $1 \times 10^5$ carboxyfluorescein succinimidyl ester (CFSE) labeled CD4$^+$/FOXP3$^{GFP-}$ Tcon and co-cultured with $1 \times 10^5$ irradiated CD4$^+$ T cell depleted splenocytes, together with 1 ug/ml of soluble anti-CD3 (145-2C11, Biolegend). Proliferation of responder Tcon was assessed by flow cytometry after 72 h of culture.

### Isolation of spinal cord infiltrating cells and flow cytometry
For analysis of cell infiltrated into the central nervous system, pooled spinal cord tissue from 4–5 mice per experimental group, was homogenized using a dounce homogenizer. Cells were then isolated by density centrifugation using a 28% Percoll solution, layered with PBS. Cells were stained according to standard protocols with the following antibodies: CD3 (17A2), CD4 (GK1.5), CD25 (PC61), FOXP3 (FKJ.16a), Tbet (4B10), RORgT (B2D), GATA3 (TWAJ; all Biolegend) using staining reagents and protocols as per the manufacturer's instructions (Supplementary Table 2). Flow cytometry was performed on an LSRII (Becton Dickinson) in the Stanford Shared FACS Facility and data analysis was done using FlowJo (Treestar).

### In vitro mAbCAR mRNA production, T cell transfection and transfer
Monoclonal antibody directed chimeric antigen receptors (mAbCAR) were created from the standard clone clone 1X9Q containing an anti-FITC scFv portion fused to murine CD28 and CD3ζ costimulatory domains. 1X9Q co-expressing HPSE and control 1X9Q mRNA were produced using T7 mScript™ Standard mRNA Production System from CellScript, following the manufacturer's protocols. T cells were transfected by electroporation using a Lonza nucleofector kit as per the manufacturer's protocol (Lonza). Briefly, 10×10$^6$ cells were washed in PBS and mixed with mAbCAR HPSE or 1X9Q mRNA, as well as the Lonza reagent, in 100 ul and electroporated using an Amaxa system. Cells were cultured overnight to allow for chimeric receptor expression. Electroporated cells were then washed once more and injected into mice intravenously ($1 \times 10^6$ to $1.5 \times 10^6$ cells/mouse). For MOG-specific mAbCAR Treg transfer, cells were incubated with a FITC-conjugated anti-MOG antibody in PBS for 30 minutes on ice and washed once before transfer. Control mice were treated with an i.v. injection with saline.

### Statistical analysis
Statistical analysis was performed using GraphPad Prism software, version 8.4.2. Average + or +/− SEM are shown when comparing multiple groups, as it indicates the variability between samples. Average +/− SD are shown when depicting the variability of data within a group. A Kolmogorov−Smirnov test was used to verify normality of samples. In samples with Gaussian distribution, an unpaired *t*-test or 1-way ANOVA was used to determine significant differences between 2 or more groups, respectively, or a 2-way ANOVA was used to identify effects of multiple parameters. In samples without Gaussian distribution, a non-parametric Mann−Whitney test, or Friedman test for scoring over time, was used to determine significant differences. A *p*-value less than 0.05 was considered statistically significant.

## Data collection and data analysis

For image analysis Imaris imaging software (Bitplane) was used. For flow cytometry analysis, FlowJo version 9 or 10 (Treestar) was used. For graphing and statistical analysis GraphPad Prism 8.4.2 was used.

## Reporting summary

Further information on research design is available in the Nature Portfolio Reporting Summary linked to this article.

## Data availability

Source data for Fig. 1c,e; 2a,c-g,i; 3a-d,f; 4a-f,h,i,l,o; 5b-f,h-l and all supplementary graphs are provided with the paper. Additional data that support the findings of this study are available from the corresponding author upon reasonable request. Source data are provided with this paper.

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

## Acknowledgements

We thank Dr. Manish Butte for critical reading of the manuscript and the Stanford Shared FACS Facility for technical assistance. This work was supported by National Institutes of Health grant R01DK046635 (GTN). National Institutes of Health grant K08DK080178-05 (PLB). National Institutes of Health grant R01DK096087-01 (PLB). National Institutes of Health grant U01AI101984 (PLB). National Institutes of Health grant R21AI133240-01A1 (PBL, HFK). National MS Society Pilot Research Grant PP-1503-03972 (PBL, HFK). CSGADP Pilot Award under NIH grant UO1AI101990 (TNW). BIRT supplement AR037296 (TNW). Stanford Uni-versity Institute for Immunity, Transplantation and Infection Young Investigator Award (HFK). Multiple Sclerosis Society of Canada Operat-ing Grant 3585 (HFK). MS Canada Discovery Research Grant 1037917 (HFK). Juvenile Diabetes Research Foundation grant nPOD 25-2010-648 (TNW). The Center for Translational Research at BRI (GTN). Helmsley Charitable Trust nPOD Award for Team Science (TNW).

## Author contributions

H.F.K., I.K., and P.L.B. conceptualized the project. H.F.K., I.K., G.K., and B.A.F. developed the methodology. Investigation was performed by H.F.K., H.A.M., I.K., G.K., B.A.F., J.K.R., C.O.M., N.N., G.B., A.H., S.Z.,

M.P.-C., and S.-W.T. Data was provided by L.E.W. and J.D.L. Resources were provided by I.V., J.-P.L., E.H.M., K.C.G., T.D.P. and L.S. Writing of original draft was done by H.F.K. Writing review & editing was done by H.F.K., H.A.M., I.K., and P.L.B. Supervision was provided from H.F.K., G.T.N., T.N.W., and P.L.B. Funding acquisition for the project was done by H.F.K., G.T.N., T.N.W., and P.L.B.

## Competing interests

The authors declare no competing interests.

## Additional information

**Supplementary information** The online version contains Supplementary Material available at

