## [Peer Review File · Nature Communications]

REVIEWER COMMENTS

Reviewer #1 (expert in multiple sclerosis and EAE):

In this manuscript, Martinez et al. describe their findings that Treg use heparanase (HPSE) to access IL-2 sequestered by heparan sulfate (HS) within the extracellular matrix of inflamed CNS tissue. HPSE^{-/-} Treg have impaired stability and function in vivo in EAE, and endowing Treg with HPSE enhances their ability to access HS-sequestered IL-2 and their tolerogenic function. This study may identify novel roles for HPSE and the ECM in immune tolerance and provide new avenues for improving Treg-based therapy of autoimmunity. This study is interesting, with certain concerns as below:

1. In the experiment to determine whether HPSE expression supports FoxP3⁺ Treg function in vitro, as shown in Fig. 5b and c, data show that only about 2% WT FoxP3⁺ Tregs were alive when co-cultured for 24 hrs with HSPG-expressing astrocytes (U87-MG) without IL-2; and only about 15% of those Tregs alive with IL-2. It is not clear why majority of Tregs died in culture of only 24 hrs, even with the optimal conditions (with HS/IL2). The same outcome occurred in Tregs cultured with CNS tissue (Fig. 5d,e). Even worse, only about 3-5% of CD25⁺FoxP3⁻ Tconv cells survived from this culture system. What is the survival rate when cells were cultured with media alone? These results are opposite to authors' interpretation that HS/IL-2 enhances Treg survival.
2. The purpose of a CAR (chimeric antigen receptor) T cell is for antigen-specific targeting. The structure of "CAR T" showing in Fig. S9 of this study is not a real CAR T, given that it did not have any antigen receptor in it, thus not being antigen-specific as CART cells are. Another problem is that the authors generated "MOG-specific Tregs" by incubating WT Tregs (isolated CD4⁺CD25⁺ T cells from naive WT CD45.1 mice) with FITC-conjugated mAb against MOG. It is most unlikely that this simple/easy procedure can make antigen-specific Tregs from WT CD4⁺CD25⁺ cells. If it is that simple, real CAR T cell technique (chimeric receptor expressing specific antigen) or MOG-TCR transgenic mice (2D2) will not be required.
3. Also, it is not clear why so-called "MOG-CAR-Treg", i.e., isolated CD4⁺CD25⁺ T cells from naive WT CD45.1 mice, incubated with FITC-conjugated mAb against MOG, failed to suppress EAE (Fig. 5l). Theoretically, the Treg function should be enhanced in these cells compared to WT Tregs, which, surprisingly, showed stronger inhibition of EAE (Fig. 5h). These "MOG-CAR-Tregs", even overexpressed with HPSE, showed minor, significant though, inhibition of EAE, at a less content than WT Tregs as shown in Fig. 5h. These results indicate an inconsistent data, or disapprove the importance of HPSE in Treg function, the major conclusion of this study.
4. In three EAE graphs (Figs. 5h, 5l and S8e), the "control" should be labeled as what the treatments were. Have the results shown in Fig. 5l been repeated?
5. In Results section, the citation of Figs. 6i-l is actually 5i-l, making it difficult to match results, figures and figure legends.

Reviewer #2 (expert in Treg homeostasis):

The manuscript entitled " FOXP3+ regulatory T cells use heparanase to access IL-2 bound to ECM in inflamed tissues" investigated the role of heparanase in Treg cells to access IL-2 sequestered by heparan sulfate within the extracellular matrix. Martinez et al. showed that Treg cells in humans and mice express a higher level of heparanase (HPSE) compared to Tconv. They also revealed that HPSE expression by Treg cell is needed to respond to IL-2 sequestered by heparan sulfate. Finally, they showed that this mechanism seems to be involved in EAE pathogenesis and propose this as a therapeutic approach using CAR-Treg cells. Given the pivotal role of Tregs in maintaining immune homeostasis and in several diseases, this study can be an important finding and can contribute to a better understanding of the fundamental aspects of Treg biology. However, the following issues need to be addressed before potential publication:

Major comments:

Figure 1: The number of replicates is too low to conclude. The authors should add more replicates and also quantify the results after heparinase treatment in Figure 1d.

Figure 2:

The proliferation results are interesting, but to be able to conclude about the role of HS-IL2 in cell proliferation. The experiments should be performed using a proliferation dye (CFSE, CTV, ...). Using this dye will allow the reader to appreciate the impact of HS-IL2 in cell proliferation.

Fig 1d-e: The results on Foxp3 expression are not strong. The authors should analyze the role of the HS-IL2 on natural Treg proliferation and induced Treg generation using different concentrations of IL-2 or HS-IL2.

The authors should also perform this analysis with sorted Treg and Tconv cells and cultivate them with IL-2 or HS-IL2. The current experiment can be explained as a bystander effect by the Tconv or by increased Induced Treg cell generation.

Figure 3:

The use of Treg and Teff transcripts from human colon tissue is not clear. The authors should use the CSF Treg and Teff or show that heparanase expression in Treg cells is also involved in Colitis.

All the HPSE expression analysis was performed after in vitro activation, what is the level of HPSE without in vitro activation?

The authors should also analyze the expression of HPSE in Treg and Tconv of the spleen or spinal cord during the active phase of EAE and compare this level to that of untreated mice.

Figure 4:

The western blot presented in Figure 4a seems to be cropped. The authors should provide the uncropped western blot.

The y-axis in Figure 4b is not clear. Can the authors clarify the use of this axis?

As mentioned in Figure 2 the authors should use a proliferation dye for these experiments.

Figure 4e, the manuscript does not mention how the p-Stat5 was performed. If this was performed using phosphoflow, the authors should add some representative dot plots in the figure.

Fig4g, the quality of the staining represented should be improved and the dot plot should be changed to a better one.

Fig4 I, the histograms represent CD3+CD4+ however based on the legend it should be a quantification of Foxp3+ cells.

Fig4j-L: the number of replicates is too low to conclude, the authors should add more replicates.

Fig 4n: the authors should also add a representative dot plot for Foxp3+ expression inside the CD45.1 and CD45.2 cells.

Figure 5:

Fig 5a-e: The authors should perform this analysis with sorted Treg and Tconv cells and cultivate them IL-2 or with the IL-2 loaded astrocytes/tissue. The current experiment can be explained as a bystander effect by the Tconv or by increased Induced Treg cell generation.

The authors should perform all these experiments with a proliferation dye as mentioned in Figure 2.

Fig5F: the authors should explain the in vitro defect in the suppressive function of HPSE^{-/-} Treg cells. This in vitro culture does not have HS or HS-related IL2. The authors should explain why Treg cells must express HPSE to suppress Tconv in this culture.

Fig5h: The authors should provide an analysis of the spinal cord immune compartment after EAE induction. The authors should analyze the T cell infiltration and their cytokine production (IFN γ , IL-17, GM-CSF, IL10) in the spinal cord after transfer of HPSE^{-/-} Treg or WT Treg.

The same type of analysis should be performed after the transfer of CAR-Treg HPSE or CAR-Treg.

Extended Figure 6:

The gating strategy provided is not using a CD3 antibody. The authors should change the gating strategy and add the CD3 staining.

The authors should add more replicates in Ex Fig 6a and perform statistical analysis.

Extended Figure 7:

Ex fig 7c: The authors should add an isotype control for the p-stat5 experiment.

Extended Figure 8:

The T-sne analysis doesn't provide any statistical analysis, the authors should perform statistical analysis and add some of these results in main figure 5 as mentioned above.

Reviewer #3 (expert in extracellular matrix proteoglycans in inflammation):

Martinez and colleagues present a series of experiments demonstrating a clear and important role for the heparan sulfate degrading enzyme heparanase in the function of regulatory T cells. These studies add significantly to previous observations of the importance of heparanase in functionality of the immune system and recent observations for the role of HS and the extracellular matrix in regulating cytokine and chemokine function.

The authors use an impressive range of experimental techniques including imaging, the EAE mouse model, ex vivo and in vitro cellular analysis and flow cytometry, western blotting and bone marrow chimeras to make a compelling case for heparanase in regulatory T cell function. This work is important to the field of immunology and more widely and represents a significant conceptual advance, however I have a few reservations detailed below. In particular I think the authors need to clarify/discuss the model they propose and how binding to HS would facilitate enhanced function of the IL-2 protein, presumably a structural change that facilitates better binding/function through its' receptor.

Specific points:

Figure 1, whilst the IL-2 and HS staining overlap is compelling can the authors speculate what the IL-2 staining that does not overlap with HS may be bound to or comment if it is thought to be intracellular? This also relates to the incomplete removal of IL-2 staining following heparanase digestion of sections. Can images in fig 1d be quantified as in a and b to better determine the changes in IL-2 and HS staining following heparanase treatment?

Figure 1, for clarity can the authors label the images in panel b as in panel a with 'EAE'. Similarly where the graph in panel c is labelled IL2 colocalisation it would help to be clear this is HS and IL-2 colocalisation.

Page 4 line 27, repetition of "deposition" Page 5 line 6-7 is "naïve of EAE" correct?

Fig 2, SEM inappropriate? If not needs correcting throughout and consistency across figures. Similarly consistent use of graph type showing individual data points and not just a bar with error. Also need to clarify the statements relating to n in each figure, does it reflect an individual mouse or technical replicate in each instance?

Fig. 2 panel a, the HS only control is confusing as the axis is labelled as IL-2 concentration but am I correct in thinking there is no IL-2 in this control? Also at the highest concentration of HS alone there does seem to be an increase in proliferation, is this correct?

Figure 2, have the authors performed bead alone controls throughout?

Does extended data 4A suggest the T51P mutation affects receptor binding too? Has this been checked?

Fig 2b uses heparin whereas 2A and C is HS, more explanation of why is needed. The text states they are the same apart from sulfation level, this is not strictly correct as there are some sequence differences.

Page 6, line 8, should read CD45 (blue) is "SHOWN".

Page 6 line 28, do the authors mean to refer to Fig. 2c here?

Fig. 2d and throughout, have the authors validated observations relating to FoxP3 using intracellular flow staining to support the reporter mouse model?

In figure 2 the authors discuss heparin bead bound T51P mutant, but is the whole point that there will be very little actually bound to the beads? Has the binding of WT and mutant IL-2 to the heparin beads been checked?

Is the lack of difference in FoxP3 MFI between WT IL-2 and T51P surprising given that the dose of mutant delivered on heparin beads will inherently be much lower?

The statement page 7 line 4 and 5 would justify validating the reporter data with intracellular flow cytometry of FoxP3.

Fig, 2h, whilst the results are compelling it would be more convincing to directly determine the amount of IL-2 or mutant on the beads.

In Fig.2e beads are labelled as heparin coated but not in h, confusing

Page 7 line 15-17, whilst subtle I think the authors should clarify that the cells are stripping the IL-2-Heparin complex from the beads and not just producing soluble IL-2, this is a subtle but important distinction.

Fig 3, consistency in plotting individual points rather than bar charts would help given the large error in 3c. Also consistency of SEM and SD

I am not sure I agree that HPSE “characterises” T reg page 10 line 9, although certainly differential between T reg and others, I would re-label the legend.

Page 9, line 36. Extended data figure 6a seems important to the manuscript and may fit better in the main figure.

Figure 3F does not appear to be mentioned in the text.

Figure 4.

The authors should compare the ability of WT and HPSE KO TREG cells to proliferate in response to HS bound IL-2 in solution as in figure 2a, this will determine the role of heparanase in liberating GAG-bound IL-2 from a surface vs any effects on the soluble IL-2/GAG complex.

Page 11, line 11, needs a reference to support the statement

Fig 4b. The authors should include the full western blot from 4b in supplemental for clarity.

The primer sequences for heparanase should be included in the paper.

Have the authors or others demonstrated human and mouse HPSE are comparable generally and in terms of their ability to degrade HS chains?

Figure legend, page 14 line 6 not sure the point on variability has been explained clearly here?

Fig 4c Why do the authors think the difference between wt and hpse KO is reduced at the highest concentration of beads?

Fig.4d, can the authors analyse Tconv production of IL2 to reinforce the reasoning behind this cell type being unaffected by HPSE KO.

Fig.4e, unclear how STAT5 is being measured, western? Flow?

Does extended Fig. 7b suggest there are differences between WT and HPSE ko in pSTAT3 MFI?

Fig 4f. Did the authors perform similar experiments to those in Fig. 4f in other tissues?

Fig. 4g why show the plots for spleen but not LN?

Fig.4i, is this duplicating the data in Fig. 4h?

Figure 5a, if the whole premise is IL-2 immobilised to HS then the authors should demonstrate that the astrocytes (U87-NG) used in culture have HS and that it regulates IL-2 immobilisation on their surface (as in figure 1)

Page 15, line 29-30, it may be helpful to expand on the tSNE plots to show which more specifically cells are altered between administration of WT and HPSE Treg to further support the EAE score data.

Fig 5b is there a deficit in % live in wt vs hpse ko at rest? If so why?

Page 16 lines 20, 22 and 29 refer to figure 6 instead of figure 5.

Page 18 line 2 should it be +/-?

Page 18 line 22, should on day x have a specified number?

CAR work, is there a difference in HPSE production when engineered to produce?

Page 16 line 27, should the reference to fig. 6K be to 6L. These need checking throughout.

Discussion

The authors need to clarify their thoughts on the mechanism involved in their findings. Is the model being proposed that T reg cells use heparinase to liberate IL-2 from HS structures to make it more bioavailable by being more soluble and presumably more able to bind to its' receptor? Figure 2 demonstrates that IL-2 bound to HS is more potent than soluble IL-2. This suggests a structural confirmation change in IL-2 by binding HS that promotes receptor binding. However the lack of effect on IL-2 receptor expressing T cells suggests this is somehow cell specific. Does this not make the argument that being bound to HS somehow promotes function, e.g. correct structural orientation for receptor binding, since being bound to HS should inhibit CTLL2 proliferation for the proposed model to be correct?

If the hypothesis is that HPSE releases a GAG chain from the matrix with IL-2 bound that together better promote T reg function but not other IL-2 receptor expressing cells then I think this could be stated more clearly.

HS binding proteins exhibit a wide diversity in the nature of their interaction with HS, given the apparent need to liberate IL-2 from the ECM that would suggest a high-affinity interaction or slow off-rate. It may help in the discussion to raise this issue and refer to any previous literature on the dynamics of the IL-2/HS interaction.

It would be helpful to include references to the recent papers on HS regulation of cytokine function and related review (see below). Comparing and/or contrasting the mechanism proposed here with these

recent studies on how HS regulated cytokine/chemokine function in other contexts would strengthen the discussion.

<https://www.nature.com/articles/s41590-023-01420-5>

<https://pubmed.ncbi.nlm.nih.gov/36640356/>

<https://www.science.org/doi/10.1126/sciimmunol.add1728?s=09>

<https://www.science.org/doi/10.1126/science.abp8964>

REVISIONS to Nature Communications manuscript NCOMMS-23-08147

We are grateful for the effort the reviewers put into strengthening our science. Below, we have addressed their critiques point-by-point with each critique in black font and our response in blue.

REVIEWER COMMENTS

Reviewer #1 (expert in multiple sclerosis and EAE):

In this manuscript, Martinez et al. describe their findings that Treg use heparanase (HPSE) to access IL-2 sequestered by heparan sulfate (HS) within the extracellular matrix of inflamed CNS tissue. HPSE^{-/-} Treg have impaired stability and function in vivo in EAE, and endowing Treg with HPSE enhances their ability to access HS-sequestered IL-2 and their tolerogenic function. This study may identify novel roles for HPSE and the ECM in immune tolerance and provide new avenues for improving Treg-based therapy of autoimmunity. This study is interesting, with certain concerns as below:

We thank the reviewer for their interest and supportive comments.

1. In the experiment to determine whether HPSE expression supports FoxP3⁺ Treg function in vitro, as shown in Fig. 5b and c, data show that only about 2% WT FoxP3⁺ Tregs were alive when co-cultured for 24 hrs with HSPG-expressing astrocytes (U87-MG) without IL-2; and only about 15% of those Tregs alive with IL-2. It is not clear why majority of Tregs died in culture of only 24 hrs, even with the optimal conditions (with HS/IL2). The same outcome occurred in Tregs cultured with CNS tissue (Fig. 5d,e). Even worse, only about 3-5% of CD25⁺FoxP3⁻ Tconv cells survived from this culture system. What is the survival rate when cells were cultured with media alone? These results are opposite to authors' interpretation that HS/IL-2 enhances Treg survival.

The data in the panels in question and elsewhere in the manuscript is presented as “% live over control” where values are shown as the percentage of live cells above that seen for 0 IU/mL IL-2 control. Therefore the value of 2% signifies 2% more cells are viable than in the unstimulated (0 IU/ml IL-2) condition, not that only 2% of cells were viable. This normalization was done to facilitate comparisons across different experimental conditions (beads, cell co-cultures, etc.) and across multiple cell types (Treg, T-cells, induced Treg) that naturally have different baselines for viability in the absence of IL-2.

To better illustrate what was done, below we show an example of the absolute percentages of live cells within the Treg populations (Left panel) and the “% live over control” normalization on the Right side. We have clarified this point in the text and have included this figure within the supplementary files.

2. The purpose of a CAR (chimeric antigen receptor) T cell is for antigen-specific targeting. The structure of “CAR T” showing in Fig. S9 of this study is not a real CAR T, given that it did not have any antigen receptor in it, thus not being antigen-specific as CART cells are. Another problem is that the authors generated “MOG-specific Tregs” by incubating WT Tregs (isolated CD4+CD25+ T cells from naive WT CD45.1 mice) with FITC-conjugated mAb against MOG. It is most unlikely that this simple/easy procedure can make antigen-specific Tregs from WT CD4+CD25+ cells. If it is that simple, real CAR T cell technique (chimeric receptor expressing specific antigen) or MOG-TCR transgenic mice (2D2) will not be required.

We agree that these are not classical CAR T-cells. To clarify what was done in this study, we endowed T-cells with a chimeric antigen receptor containing CD28, CD3 ζ , as well as HPSE. A schematic is shown below and is provided in supplementary figure 13a,b. The antigen specificity of the CAR is against fluorescein isothiocyanate (FITC). Therefore, specificity can be established using any FITC-conjugated antibody. This construct was selected based on our previously published work (PMID: 29046484) where we demonstrated that cells engineered in this manner are functional.

We previously called this version of CAR T-cells “antibody directed CAR” or “mAbCAR”, To address the reviewer’s comment and prevent confusion, we have switched to this nomenclature herein the text.

3. Also, it is not clear why so-called “MOG-CAR-Treg”, i.e., isolated CD4+CD25+ T cells from naive WT CD45.1 mice, incubated with FITC-conjugated mAb against MOG, failed to suppress EAE (Fig. 5i). Theoretically, the Treg function should be enhanced in these cells compared to WT Tregs, which, surprisingly, showed stronger inhibition of EAE (Fig. 5h). These “MOG-CAR-Tregs”, even overexpressed with HPSE, showed minor, significant though, inhibition of EAE, at a less content than WT Tregs as shown in Fig. 5h. These results indicate an inconsistent data, or disapprove the importance of HPSE in Treg function, the major conclusion of this study.

We respectfully offer that the experiments in question are not inconsistent but rather that the two experiments involve different cell types and different experimental protocols, making direct comparison between them impossible. In particular, the experiment in Figure 5h involves natural Treg that were transferred before the induction of EAE. Conversely, in Figure 5l mAbCAR Treg were transferred after the onset of disease. This latter approach is more challenging - indeed most protocols transfer Treg before disease initiation (PMID: 12391178, PMID: 29857928). This was done to set a high bar for evaluating the impact of HPSE expression on mAbCAR Treg.

These protocol distinctions have been made clear in the text (page 16, line 28). In addition, we have added arrows to the disease course graphs to clarify when during the onset of EAE the mAbCAR Treg were added (this information is included in the schematic in Supplementary Fig 11g as well).

4. In three EAE graphs (Figs. 5h, 5l and S8e), the "control" should be labeled as what the treatments were. Have the results shown in Fig. 5l been repeated?

Thank you for raising this point. The control treatment for these experiments were saline injections. We have clarified this in the figures (Fig. 5h and l), in the figure legend (page 18 lines 18 and 30) and in the methods (page 10 line 4 and page 11 line 11). The experiment shown in Fig.5l was performed twice.

5. In Results section, the citation of Figs. 6i-l is actually 5i-l, making it difficult to match results, figures and figure legends.

These changes have been made, thank you for bringing them to our attention.

Reviewer #2 (expert in Treg homeostasis):

The manuscript entitled " FOXP3+ regulatory T cells use heparanase to access IL-2 bound to ECM in inflamed tissues" investigated the role of heparanase in Treg cells to access IL-2 sequestered by heparan sulfate within the extracellular matrix. Martinez et al. showed that Treg cells in humans and mice express a higher level of heparanase (HPSE) compared to Tconv. They also revealed that HPSE expression by Treg cell is needed to respond to IL-2 sequestered by heparan sulfate. Finally, they showed that this mechanism seems to be involved in EAE pathogenesis and propose this as a therapeutic approach using CAR-Treg cells. Given the pivotal role of Tregs in maintaining immune homeostasis and in several diseases, this study can be an important finding and can contribute to a better understanding of the fundamental aspects of Treg biology. However, the following issues need to be addressed before potential publication:

We thank the reviewer for their supportive comments.

Major comments:

6. Figure 1: The number of replicates is too low to conclude. The authors should add more replicates and also quantify the results after heparinase treatment in Figure 1d.

We performed immunofluorescent staining on mice from 3 independent EAE experiments, with very similar results (compare Fig. 1a and 1b with an example of staining from a different EAE experiment shown below (cerebellar tissue harvest at day 25 post immunization, and control tissue)). We performed Imaris rendering and quantification on 4 mice from one experiment, to prevent inter-experiment variability, and quantified 3-5 different locations for each area of interest (lesion, peri-lesion and pia) from these animals. We feel that the comparison of multiple time points over the course of EAE provides sufficient support for the notion that IL-2 is increasingly associated with HS as the disease progresses. We have also added a video of another 3D-rendered spinal cord lesion as Supplementary Movie 1 to illustrate the abundance and overlap of IL-2 and HS in inflamed CNS tissue.

We have quantified IL-2 binding within the heparinase-treated tissue and have added these data as Figure 1e (see below).

IL2 colocalization Hepase treatment

Figure 2:

7. The proliferation results are interesting, but to be able to conclude about the role of HS-II2 in cell proliferation. The experiments should be performed using a proliferation dye (CFSE, CTV, ...). Using this dye will allow the reader to appreciate the impact of HS-IL2 in cell proliferation.

Given that natural Tregs are minimally proliferative *in vitro* (PMID: 9885918, PMID: 967004) we understand that the reviewer is referring to the experiments with CTLL2 cells in Figure 2. Unfortunately, in our hands CTLL2 cells labeled with CFSE or other proliferative dyes exhibit only a single peak of proliferation (see below). Our understanding of this is that because CTLL2 cells are a cell line, they proliferate in unison or die depending on IL-2 availability. For this reason resazurin or other similar quantification tools are typically used to assess CTLL2 proliferation.

Nonetheless, we appreciate that it is important to provide an orthogonal method of assessing CTLL2 proliferation. To this end, we have used tritiated thymidine ($[^3\text{H}]\text{TdR}$) to quantify plate-bound HS effects on IL-2-mediated proliferation. The incorporation of ($[^3\text{H}]\text{TdR}$), quantified as counts per minute (CPM) is a well-established method of assessing cell numbers.

8. Fig 1d-e: The results on Foxp3 expression are not strong. The authors should analyze the role of the HS-IL2 on natural Treg proliferation and induced Treg generation using different concentrations of IL-2 or HS-IL2.

To address the reviewers' question with regards to induced Treg generation, we have induced Foxp3 Tregs in the absence or presence of HS. In particular, we stimulated CD4+ T cells with anti-CD3/anti-CD28 in the presence of 50 ng/ml TGF-beta and a suboptimal dose of 20 IU/ml or 100 IU/ml IL-2 preincubated with or without HS. We observed significantly increased frequency and viability of Foxp3+ Treg by the preincubation of IL-2 with HS. Those data are now included in Figure 2d (previously this figure only included the viability data of these experiments, we have now included the frequency of Foxp3-expressing cells after induction).

Regarding natural Treg proliferation, it is well-established that natural Tregs are minimally proliferative *in vitro* (PMID: 9885918, PMID: 967004). We understand the reviewer is asking us to enrich natural Tregs and perform a proliferation assay in a similar manner to that performed with the CTLL-2s. We performed two independent experiments to measure this and we see minimal proliferation. We have provided an example plot of this below.

9. The authors should also perform this analysis with sorted Treg and Tconv cells and cultivate them with IL-2 or HS-IL2. The current experiment can be explained as a bystander effect by the Tconv or by increased Induced Treg cell generation.

The data in Figure 2f and g, showing capacity of cells to access heparin-bead-bound IL-2, was performed with enriched Treg cultures, which were magnetic bead-sorted based on CD25 expression, generally giving us a ~75% frequency of Foxp3+ Treg among the preparations. We have clarified this in the methods and results section.

To address this question we used CD25+-enriched Treg and cultured this fraction, as well as the CD4+ CD25- Tconv fraction on plates that were coated with or without HS and pre-

incubated with IL-2. The residual IL-2 not bound to HS was then washed off before cells were added. We then assessed the viability of Foxp3⁺ Treg, in both the Treg (CD25⁺) and Tconv (CD25⁻) enriched fraction. We observed that the survival of enriched Treg was supported by plate-bound HS-IL2, again supporting a role for HS-bound IL-2 in supporting Treg viability. In contrast, the survival of Treg present at low frequency in the Tconv enriched fraction was not significantly supported by HS-IL2, ruling out a bystander effect of Tconv on Treg.

In addition, we also assessed the viability of FACS-sorted Foxp3⁺ Treg, cultured on plate-bound HS-IL2 and observed that over 24 and 48 hours plate-bound HS-IL2 was able to support Treg survival, showing a definitive role for HS-bound IL-2 in supporting Treg viability.

With regard to a potential effect on Treg induction under the conditions shown in Figure 2e and f, Treg induction should not occur in the absence of TGF-beta. To this point, we did not see an increase of Foxp3⁺ T cells when cells were cultured with either soluble or bead-bound

IL2, indicating that heparin-bead-IL2 does not promote Treg expansion. We have added this data to Supplementary Figure 6c. Likewise, we did not see an increase of Foxp3+ T cell numbers when CD25- Tconv were cultured on plate-bound HS/IL-2, as shown above.

Figure 3:

9 .The use of Treg and Teff transcripts from human colon tissue is not clear. The authors should use the CSF Treg and Teff or show that heparanase expression in Treg cells is also involved in Colitis.

To clarify, this is a dataset containing colon tissue from a registry of inflammatory gastrointestinal disease patients, including the autoimmune condition Crohn's Disease and Ulcerative Colitis. We used this dataset to demonstrate that across human inflammatory conditions, Treg express higher levels of HPSE than Tconv. We have clarified this in the text (Page 9 line 35) and added the data for the different disease groups (and healthy controls) in Supplementary Fig. 8e.

We thank the reviewer for the suggestion. However, it is not possible to isolate and purify sufficient numbers of Treg from CSF to culture in vitro and nowhere near enough to treat colitis in mice. We have performed this model previously (PMID: 21518860) and transfer of 2×10^5 Treg are required. It is simply not possible to collect this many Treg from the CSF of even multiple mice.

10. All the HPSE expression analysis was performed after in vitro activation, what is the level of HPSE without in vitro activation?

Unactivated Treg express low amounts of HPSE at baseline - this is shown in Fig. 3b.

11. The authors should also analyze the expression of HPSE in Treg and Tconv of the spleen or spinal cord during the active phase of EAE and compare this level to that of untreated mice.

We agree that analyzing the expression HPSE in Treg & Tconv in EAE, especially at the site of autoimmune inflammation i.e. the spinal cord, would be very informative. We have performed the assay that the reviewer requested. However,; there are simply too few resident Treg present to say anything meaningful about HPSE expression on these cells. Those data are included below.

Figure 4:

10. The western blot presented in Figure 4a seems to be cropped. The authors should provide the uncropped western blot.

We have provided the uncropped blots in Supplementary Figure 9. We have also provided uncropped blots for the data shown in Figure 3b in Supplementary Figure 7.

11. The y-axis in Figure 4b is not clear. Can the authors clarify the use of this axis?

The Y-axis in Figure 4b is expressed as “IL-2 equivalent dose” because in this experiment we are using recombinant versions of IL-2 (either recombinant wild-type IL-2 or recombinant T51P IL-2). Because these reagents might have different bioactivity, we standardize their functional activity to a standard curve of commercial IL-2. This was done in Supplementary Figure 4a using CTLL2 cells. An example of how this equivalent dose is calculated (interpolated from the standard curve) is shown below.

This is similar to the way the effective dose of commercial preparations of IL-2 are determined (https://www.rndsystems.com/products/recombinant-human-il-2-protein-cf_bt-002, <https://www.biolegend.com/en-ie/products/recombinant-mouse-il-2-carrier-free-4526>).

12. As mentioned in Figure 2 the authors should use a proliferation dye for these experiments.

Our understanding is that the reviewer is again referring to the proliferative studies with CTLL2 cells in Figure 4b. As noted above, cellular proliferation in a synchronous cell line like the CTLL-2 will not yield traditional peaks seen with primary cells stained with CFSE or CTV. Therefore, resazurin is typically used to quantify CTLL2 proliferation.

13. Figure 4e, the manuscript does not mention how the p-Stat5 was performed. If this was performed using phosphoflow, the authors should add some representative dot plots in the figure.

pSTAT5 was performed by phosphoflow. We have added this information in the figure legend and we have provided example flow plots of soluble IL2 and plate-bound HS-IL2 stimulation, as well as FMO and unstimulated (4 C) controls (Supplementary Fig. 10b).

14. Fig4g, the quality of the staining represented should be improved and the dot plot should be changed to a better one.

We have exchanged the dot plot with higher resolution ones, to show an example of the staining (Fig. 4g).

15. Fig4 I, the histograms represent CD3+CD4+ however based on the legend it should be a quantification of Foxp3+ cells.

We have changed the y-axis labels of the graphs to better depict the data shown (% Foxp3+ cells among CD3+/CD4+ cells).

16. Fig4j-L: the number of replicates is too low to conclude, the authors should add more replicates.

These mixed bone marrow chimera experiments were performed twice with three individual mice 3, yielding statistically significant results. Data are shown from three mice in one representative experiment.

17. Fig 4n: the authors should also add a representative dot plot for Foxp3+ expression inside the CD45.1 and CD45.2 cells.

We have added a representative dot plot for the FOXP3 expression in CD45.1 and CD45.2 cells, as well as a gating scheme and the quantification of Foxp3+ Treg frequencies among the transferred cells in Supplementary Figure 10h and i.

Figure 5:

18. Fig 5a-e: The authors should perform this analysis with sorted Treg and Tconv cells and cultivate them IL-2 or with the IL-2 loaded astrocytes/tissue. The current experiment can be explained as a bystander effect by the Tconv or by increased Induced Treg cell generation. The authors should perform all these experiments with a proliferation dye as mentioned in Figure 2.

We confirmed that there is no effect of HS-IL2 on Tconv or Treg induction using our bead assay and plate-bound HS-IL2 (see item #9). Given that was the case with soluble reagents, we would not expect a different answer with IL-2 bound to cells. Nonetheless, to investigate whether T-cells can access IL-2 bound within CNS tissue and whether this is HPSE dependent, we first pre-incubated spinal cord tissue with IL-2 or T51P IL-2. We then used CTLL2 cells that were engineered to express HPSE and added these to the spinal cord/IL-2 cultures. We find that conventional CTLL2 cells were unable to access tissue bound IL-2 but that expression of HPSE allowed them to do so. These data, provided in Supplementary Figure 11e and f, demonstrate that T-cells sensitive to IL-2 access IL-2 bound to HS in CNS tissue.

CTLL2 cells that overexpress HPSE can access IL-2 sequestered by CNS tissue. CTLL2 cells were modified to overexpress HPSE. These were then incubated in spinal cord tissue pre-incubated in IL-2 or T51P IL-2. Proliferation was assessed using resazurin. Only wildtype IL-2 promoted CTLL2 proliferation. These data demonstrate that HS-bound IL-2 in CNS can support T-cell proliferation.

19. Fig5F: the authors should explain the in vitro defect in the suppressive function of HPSE^{-/-} Treg cells. This in vitro culture does not have HS or HS-related IL2. The authors should explain why Treg cells must express HPSE to suppress Tconv in this culture.

We have shown in Figure 2h and 2i that Tregs have the capacity to “strip” IL-2 from beads and in Figure 5a that Treg can strip this from other cells while HPSE^{-/-} Treg could not. HS polymers are known to decorate the surface of T-cells (syndecan 1-4, glypicans). We propose that HPSE^{-/-} Tregs cannot “strip” the bound IL-2 from other cells, failing to suppress.

20. Fig5h: The authors should provide an analysis of the spinal cord immune compartment after EAE induction. The authors should analyze the T cell infiltration and their cytokine production (IFN γ , IL-17, GM-CSF, IL10) in the spinal cord after transfer of HPSE^{-/-} Treg or WT Treg.

The same type of analysis should be performed after the transfer of CAR-Treg HPSE or CAR-Treg.

To address this question, we quantified T helper subsets in pooled spinal cord infiltrates collected from either recovering or continuously sick mice (3 mice per group). To identify T-helper subsets, we used the on transcription factors Tbet for Th1 cells, Gata3 for Th2 cells, and RORgt for Th17 cell, and Foxp3, which would represent both donor and recipient Treg, and the Foxp3-reporter GFP in donor-derived Treg. Below is the quantification of our transferred cells and staining of the canonical T helper transcription factors. We have added this data to Supplementary Figure 12.

Extended Figure 6:

21. The gating strategy provided is not using a CD3 antibody. The authors should change the gating strategy and add the CD3 staining.

The cells in question were isolated from over 20 surgical resection cases involving human donors. Cell isolation and sorting were performed soon after colon resection. Unfortunately, it is not possible to add an additional marker to the staining panel for these cells at this time, given that the samples were previously collected and analyzed.

The reason anti-CD3 antibodies were not used to sort cells at that time was that a subset of cells were subsequently activated using anti-CD3 and there was a concern multiple antibodies targeting this receptor could adversely impact activation. Instead, a panel of other

markers (CD11c-/CD219-/CD4+/CD25+/CD127-) was used to identify T-reg. To our knowledge, no other cell population has this exact profile.

22. The authors should add more replicates in Ex Fig 6a and perform statistical analysis.

The data in question represent pooled cells from multiple animals (n= 4-5 per condition for a total of 30 animals). This large number of animals was needed to isolate sufficient numbers of Treg for culture, activation, and mRNA harvest and sequencing. We did not also perform triplicate replicates as suggested by the reviewer because the necessary number of animals (~90) would be impossible to harvest, sort, and analyze. Nonetheless, the large number of animals involved should account for the biological variability in these data.

Extended Figure 7:

23. Ex fig 7c: The authors should add an isotype control for the p-stat5 experiment.

We included a fluorescence minus one (FMO) control for our pSTAT5 experiments to identify gating boundaries and an unstimulated control for background correction. We typically do not use isotype controls to determine positive versus negative cells or to set gates. While there is no universal consensus regarding the use of isotype controls, our approach has broad support in the literature (PMID: [32926370](https://pubmed.ncbi.nlm.nih.gov/32926370/); <https://www.bio-rad-antibodies.com/flow-cytometry-guide-isotype-controls.html>).

Extended Figure 8:

24 .The T-sne analysis doesn't provide any statistical analysis, the authors should perform statistical analysis and add some of these results in main figure 5 as mentioned above.

To address the reviewer's concern and given that they also asked for detailed flow cytometry of T-cells in the spinal cord (see item #20), we have removed the tSNE plots in question.

Reviewer #3 (expert in extracellular matrix proteoglycans in inflammation):

Martinez and colleagues present a series of experiments demonstrating a clear and important role for the heparan sulfate degrading enzyme heparanase in the function of regulatory T cells. These studies add significantly to previous observations of the importance of heparanase in functionality of the immune system and recent observations for the role of HS and the extracellular matrix in regulating cytokine and chemokine function.

The authors use an impressive range of experimental techniques including imaging, the EAE mouse model, ex vivo and in vitro cellular analysis and flow cytometry, western blotting and bone marrow chimeras to make a compelling case for heparanase in regulatory T cell function.

This work is important to the field of immunology and more widely and represents a significant conceptual advance, however I have a few reservations detailed below. In particular I think the authors need to clarify/discuss the model they propose and how binding to HS would facilitate enhanced function of the IL-2 protein, presumably a structural change that facilitates better binding/function through its' receptor.

We thank the reviewer for their supportive comments and their input.

Specific points:

25. Figure 1, whilst the IL-2 and HS staining overlap is compelling can the authors speculate what the IL-2 staining that does not overlap with HS may be bound to or comment if it is thought to be intracellular? This also relates to the incomplete removal of IL-2 staining following heparanase digestion of sections. Can images in fig 1d be quantified as in a and b to better determine the changes in IL-2 and HS staining following heparanase treatment?

The IL-2 staining that does not overlap with HS could be bound to its receptor complex. Additionally, we could be staining IL-2 that is being turned over bound to HS-proteoglycans or IL2R complex. Finally, we could be staining IL-2 that is being made *de novo*. We have quantified the colocalization of IL-2 with HS post heparanase treatment and included it in figure 1, panel e. After treatment, any remaining IL-2 is mostly not colocalized with HS, any remaining HS (very little) is still associated with IL2.

IL2 colocalization Heparanase treatment HS colocalization Heparanase treatment

26. Figure 1, for clarity can the authors label the images in panel b as in panel a with 'EAE'. Similarly where the graph in panel c is labeled IL2 colocalisation it would help to be clear this is HS and IL-2 colocalisation.

We have added EAE to panels b and d for clarification and have labeled the colocalization data in panels c and e with "IL2 colocalization with HS".

27. Page 4 line 27, repetition of "deposition"

This has been corrected.

28. Page 5 line 6-7 is "naïve of EAE" correct?

Typo, thanks. Supposed to say 'naive and EAE'.

29. Fig 2, SEM inappropriate? If not needs correcting throughout and consistency across figures. Similarly consistent use of graph type showing individual data points and not just a bar with error. Also need to clarify the statements relating to n in each figure, does it reflect an individual mouse or technical replicate in each instance?

The consensus we used for showing individual data points or only bars is the following: whenever replicates consisted of individual mice we show bars for the calculated average, as well as the individual data points, whereas for technical replicates such as multiple wells, we only show bars. The n are mentioned throughout the legends. We consistently use SEM when comparing means, because this conveys the likelihood that 2 means are different. We use SD when describing the variability of data within a group (for example of different patients). We hope this clarifies our use within the manuscript and we have added this information to the methods section.

30. Fig. 2 panel a, the HS only control is confusing as the axis is labelled as IL-2 concentration but am I correct in thinking there is no IL-2 in this control? Also at the highest concentration of HS alone there does seem to be an increase in proliferation, is this correct?

The HS only control is indeed not pre-incubated with IL-2, nor is IL-2 added to this condition. To serve as a proper control for HS-bound IL-2 it was prepared at the same concentration as the HS present in the pre-co-incubated HS-IL2. For HS/IL-2, a stock of 1 mg/ml HS and 1×10^6 U/ml IL-2 was co-incubated and further diluted for the different doses. For the HS only control, a stock of 1 mg/ml HS was made and diluted in the same manner. We have clarified this in the methods and figure legend.

31. Figure 2, have the authors performed bead alone controls throughout?

Yes, we included bead only controls throughout. In general, we observe negligible effects of beads alone. One example is given below, where the survival of Treg with beads only is only 2% over that of Treg cultured in media without IL-2 (grey bar), and similar to beads incubated with 1×10^2 IU/ml of the T51P mutant. To prevent confusion, we have omitted these controls from the figures.

32. Does Supplementary 4A suggest the T51P mutation affects receptor binding too? Has this been checked?

We cannot answer that accurately from Supplementary figure 4a as we are monitoring proliferation of CTLL-2 cells, which measures bioactivity. We did not measure these parameters ourselves, but previous work has shown that the T51P mutation has a 10-fold lower receptor binding affinity. However the bioactivity of the mutant was nearly identical to wildtype IL-2 (same as our results) due to impacts on receptor internalization (PMID: 8662876).

33. Fig 2b uses heparin whereas 2A and C is HS, more explanation of why is needed. The text states they are the same apart from sulfation level, this is not strictly correct as there are some sequence differences.

The reason for this is that we required the thiol group for thiol-epoxy “click” chemistry reactions to covalently link heparin to magnetic beads. We have added a comment to clarify this point in the manuscript. Please see page 6 lines 27.

34. Page 6, line 8, should read CD45 (blue) is “SHOWN”.

Typo, thank you.

35. Page 6 line 28, do the authors mean to refer to Fig. 2c here?

Yes, typo. Thank you.

36. Fig. 2d and throughout, have the authors validated observations relating to FoxP3 using intracellular flow staining to support the reporter mouse model?

We understand that the reviewer questions whether GFP expression is truly indicative of Foxp3 expression in the transgenic reporter mice. For this, we refer to the original study that described these mice, developed by the group of Dr. Alexander Rudensky: PMID: 15780990. The FOXP3.eGFP mice were generated by inserting the complete eGFP coding sequence in-frame into the first coding exon of the Foxp3 gene, yielding an allele that encodes a chimeric GFP-Foxp3 fusion protein (Foxp3^{gfp}). Therefore, any cell that is GFP⁺ also expresses Foxp3. Thorough characterization of this mouse showed that the Foxp3^{gfp} allele fully recapitulates Foxp3 function, showing comparable Foxp3⁺ Treg frequencies and suppressive function, compared to wt mice.

37. In figure 2 the authors discuss heparin bead bound T51P mutant, but is the whole point that there will be very little actually bound to the beads? Has the binding of WT and mutant IL-2 to the heparin beads been checked?

We addressed this issue in extended figure 4c which is an indirect measurement of binding. Others have examined the binding of the T51P mutant to heparin elsewhere (PMID: 9597549).

38. Is the lack of difference in FoxP3 MFI between WT IL-2 and T51P surprising given that the dose of mutant delivered on heparin beads will inherently be much lower?

The reviewer is correct that the amount of T51P bound to the beads is lower. However, it is not completely absent. In extended figure 4c we quantify equivalent units of soluble IL-2 delivered by the heparin coated beads (using CTLL2 proliferation as a readout). In the lowest example of beads used (incubated with 1×10^2 cytokine), the equivalent dose of bead-bound T51P is comparable to 6 U/ml of IL-2 and that of bead-bound WT IL-2 comparable to 17 U/ml. These data suggest both levels of IL-2 are sufficient to induce Foxp3 expression in combination with T cell stimulation.

39. The statement page 7 line 4 and 5 would justify validating the reporter data with intracellular flow cytometry of FoxP3.

We have validated Foxp3 staining to confirm the Foxp3.GFP signal. As before (see item #36), we reference again the reporter mouse design wherein GFP is fused to Foxp3 (PMID: 15780990).

40. Fig, 2h, whilst the results are compelling it would be more convincing to directly determine the amount of IL-2 or mutant on the beads.

To address this we performed IL-2 staining on these beads after co-incubation with either Tconv or Treg. We observe that only Treg reduces the amount of IL-2 signal.

41. In Fig.2e beads are labeled as heparin coated but not in h, confusing

Thank you for pointing out this discrepancy. We have added heparin to Fig. 2h.

42. Page 7 line 15-17, whilst subtle I think the authors should clarify that the cells are stripping the IL-2-Heparin complex from the beads and not just producing soluble IL-2, this is a subtle but important distinction.

Thank you, we agree and have a sentence saying just that. See page 7 line 20.

43. Fig 3, consistency in plotting individual points rather than bar charts would help given the large error in 3c. Also consistency of SEM and SD

As described above (see item #29), when using technical replicates, we do not show individual points. When we use individual mice/human samples, however, we indicate this by adding individual points on the graph. We consistently use SEM when comparing means, because it is a measure of variability between samples and conveys the likelihood that 2 means are different. We use SD when describing the variability of data within a group (for example of different patients). We hope this clarifies our use within the manuscript and we have added this information to the methods section.

44. I am not sure I agree that HPSE “characterises” T reg page 10 line 9, although certainly differential between T reg and others, I would re-label the legend.

We agree with this point, we have changed the legend. Please see page 10 line 8 for change.

45. Page 9, line 36. Supplementary figure 6a seems important to the manuscript and may fit better in the main figure.

We placed this file in the Supplementary figure 8a because this plot depicts pooled animals ($n \geq 10$) for sufficient Treg numbers. Thus we only have a single data point depicting HPSE expression after IL-2 exposure for 24 and 72 hours.

46. Figure 3F does not appear to be mentioned in the text.

This is a good catch, thank you for pointing this out. We will reference this figure in the manuscript.

Figure 4.

47. The authors should compare the ability of WT and HPSE KO TREG cells to proliferate in response to HS bound IL-2 in solution as in figure 2a, this will determine the role of heparanase in liberating GAG-bound IL-2 from a surface vs any effects on the soluble IL-2/GAG complex.

We performed an experiment with Treg proliferation, but these cells do not proliferate in vitro in response to IL-2, unlike the CTLL-2 cells.

48. Page 11, line 11, needs a reference to support the statement

We have added a reference (PMID: 29728511) to the statement, thanks. Please see page 11, line 12.

49. Fig 4b. The authors should include the full western blot from 4b in supplementary for clarity.

We assume the reviewer is referring to figure 4a. We have included all full western blot scans in Supplementary Figures 7 and 9.

50. The primer sequences for heparanase should be included in the paper.

We have included these in the manuscript under the methods section within the real-time PCR section, thank you. Supplementary page 7 line 11-15.

51. Have the authors or others demonstrated human and mouse HPSE are comparable generally and in terms of their ability to degrade HS chains?

We have not demonstrated that human and mouse HPSE are comparable but protein alignment suggests the two species share ~75% homology. We also attempted to show activity with primary cells, but technically it was not feasible to show, with only purified enzyme showing activity (see below; increasing signal overtime indicates cleavage by HPSE into fluorogenic species).

52. Figure legend, page 14 line 6 not sure the point on variability has been explained clearly here?

We see that the language in the legend was not very clear and incorrect. We have corrected this. To clarify, we used a two-way ANOVA test to determine the source of the variation in the data: the genotype, or the dose of the bead-delivered IL-2. Because only the variation driven by the genotype is meaningful, we only show the p-value of this factor.

53. Fig 4c Why do the authors think the difference between wt and hpse KO is reduced at the highest concentration of beads?

We cannot speak to why that might be exactly, but to speculate perhaps that concentration of IL-2 is maximal for both HPSE^{-/-} and WT Treg to persist without reliance on HPSE. IL-2 can bind on and off the beads, and at this concentration diffusion might be sufficient to sustain the Tregs. As a consequence, the difference between the ability of HPSE^{-/-} and WT Treg to access sequestered IL-2 will be most prominent at limited concentrations.

54. Fig.4d, can the authors analyse Tconv production of IL2 to reinforce the reasoning behind this cell type being unaffected by HPSE KO.

It is well-established that conventional T cells produce endogenous IL-2 (PMID: 18062768). Therefore, they would not be expected to rely on external sources of IL-2

55. Fig.4e, unclear how STAT5 is being measured, western? Flow?

STAT5 is measured by flow cytometry. We explain this in our supplementary methods section under flow cytometry page 5 line 24 to page 6 line 3. We have added this clarification to the figure 4e legend, see manuscript page 14 line 12.

56. Does extended Fig. 7b suggest there are differences between WT and HPSE ko in pSTAT3 MFI?

We assume the reviewer means pSTAT5. It does suggest that HPSE^{-/-} have different pSTAT5 kinetics. This could be from failure to cleave IL-2 passively sticking to HS-proteoglycans, hence why there isn't a difference in initial kinetics when IL-2 is first added but later on.

57. Fig 4f. Did the authors perform similar experiments to those in Fig. 4f in other tissues?

Yes, we did perform similar experiments in other tissues but only in WT mice and compared Treg vs Tcon pSTAT5 intensity. Shown below, we see the level of pSTAT5 is higher amongst Tregs than Tcons, except in blood. We find this result interesting because HS/IL-2 complexes are not present in the blood, but would be present in the mesenteric LN, spleen and thymus, lymphoid organs where IL-2 is primarily produced (PMID: 29728511).

58. Fig. 4g why show the plots for spleen but not LN?

We selected one to be shown because staining in either spleen or lymph node yields a similar staining pattern and felt just one organ was sufficient for a representative plot.

59. Fig.4i, is this duplicating the data in Fig. 4h?

Fig. 4h. Showed percentage of FOXP3+ Treg among CD4+ T cells in the spleens (lefts panel) and inguinal lymph nodes (right panel) of a cohort of adult (3 to 6-month-old) wt and HPSE^{-/-} mice. Fig. 4i. was the quantification of FOXP3+ Treg frequencies among CD4+ T cells in lymphoid (left panel) and non-lymphoid (right panel) tissues of wt and HPSE^{-/-} mice within a smaller cohort of 8-12 week old mice. BM, bone marrow; LI, large intestine. As the data in panel 4h and part of panel 4i was indeed a bit replicative, we have moved panel 4h to Supplementary Fig. 10e, and exchanged it with the data showing Treg frequencies across different ages of wt and HPSE^{-/-} mice, which we think is more informative (this data was shown in Extended Fig. 10g before).

60. Figure 5a, if the whole premise is IL-2 immobilised to HS then the authors should demonstrate that the astrocytes (U87-NG) used in culture have HS and that it regulates IL-2 immobilisation on their surface (as in figure 1)

It has been shown before that astrocytes, including the U87-MG cell line, express HS abundantly (PMID: 18279312, PMID: 25233401). In addition, when we treat IL-2 loaded astrocytes with heparinase, their IL-2-dependent support of Treg survival is reduced (see below). We have added this data in Supplementary Figure 8.

61. Page 15, line 29-30, it may be helpful to expand on the tSNE plots to show which more specifically cells are altered between administration of WT and HPSE Treg to further support the EAE score data.

To address this point and to provide quantitative more measurements of these differences we have generated a more simple data plot, showing equal numbers of CD45⁺ cells in the spinal cord of saline-treated (Ctl), and WT or HPSE^{-/-} Treg treated mice, see below. These data show that Foxp3⁺ Treg numbers are present at a higher frequency in the spinal cord of mice that received WT Tregs

62. Fig 5b is there a deficit in % live in wt vs hpse ko at rest? If so why?

No, there is not a difference in live cells at baseline between WT and HPSE KO Tregs.

63. Page 16 lines 20, 22 and 29 refer to figure 6 instead of figure 5.

Thank you for catching this. We changed all mentions of figure 6 to figure 5.

64. Page 18 line 2 should it be +/-?

Yes, that was a typo - thank you for catching that!

65. Page 18 line 22, should on day x have a specified number?

Thank you for pointing out this important omission. In this experiment, the cells were transferred on day 14 after induction, after onset of symptoms. We have changed this in the manuscript, please see page 18 line 29.

66. CAR work, is there a difference in HPSE production when engineered to produce?

To evaluate whether the CAR construct in question increased HPSE we evaluated gene expression in T-cells transduced with this, as shown below.

67. Page 16 line 27, should the reference to fig. 6K be to 6L. These need checking throughout.

Thank you for catching this. We mistakenly kept references to figure 6 from an old version and didn't update to figure 5. We have changed these issues in the manuscript for figure 5 and will carefully check the other figures too. Please see corrections on page 16-17.

Discussion

68. The authors need to clarify their thoughts on the mechanism involved in their findings. Is the model being proposed that T reg cells use heparinase to liberate IL-2 from HS structures to make it more bioavailable by being more soluble and presumably more able to bind to its receptor? Figure 2 demonstrates that IL-2 bound to HS is more potent than soluble IL-2. This suggests a structural confirmation change in IL-2 by binding HS that promotes receptor binding. However the lack of effect on IL-2 receptor expressing T cells suggests this is somehow cell specific. Does this not make the argument that being bound to HS somehow promotes function, e.g. correct structural orientation for receptor binding, since being bound to HS should inhibit CTLL2 proliferation for the proposed model to be correct?

If the hypothesis is that HPSE releases a GAG chain from the matrix with IL-2 bound that together better promote T reg function but not other IL-2 receptor expressing cells then I think this could be stated more clearly.

HS binding proteins exhibit a wide diversity in the nature of their interaction with HS, given the apparent need to liberate IL-2 from the ECM that would suggest a high-affinity interaction or slow off-rate. It may help in the discussion to raise this issue and refer to any previous literature on the dynamics of the IL-2/HS interaction.

We have performed modeling of a heparin tetrasaccharide binding to IL-2 and then seeing where that site aligns when the heparin/IL-2 is bound to the IL2R complex (see below). The highest scoring site (a site defined by a discontinuous cluster of basic amino acids) on the face of IL-2 does not appear to interfere with the receptor binding to IL-2 when heparin (or HS) is present. However molecular docking of the heparin tetrasaccharide (~1kDa molecular weight (MW), PMID: 28007564) does not necessarily reflect the potential molecular weights of heparin/HS found within tissues (HS: ~30 kDa, heparin: ~15 kDa, PMID: 22566225). Thus, while considering steric hindrance, multiple IL-2 molecules may be captured by heparin and more so with HS. With this knowledge we feel like there are multiple effects occurring. First, IL-2 is captured through electrostatic based interactions, with an estimated binding affinity being ~0.5 μ M for HS (PMID: 9417813). However avidity (synonymous with 'functional affinity') effects will allow reacquisition of IL-2 during on and off binding events (not quantified here). Concurrent with IL-2 binding to HS, heparanase (HPSE) on the cell surface captures the HS/IL-2 complex and either holds it near the surface or liberates it. Now high affinity IL2R alpha (CD25) displays an affinity for IL-2 ~50x stronger at ~ 10 nM than HS (PMID: 20732639) and will consequently "capture" the HS/IL-2 liberated or bound by HPSE. Therefore, we feel the predominant role of HS/IL-2 and HPSE is to locally enhance the concentration of IL-2 for acquisition by the IL2R complex and initiate signaling.

Additional support for this idea comes from work with a heparin mimetic and inhibitor of HPSE, PI-88 (PMID: 15196816). When we coat HS on a plate and incubate with IL-2 we see CTLL-2 proliferation. However, if we treat with PI-88 the cells display decreased growth in response to IL-2, the only factor provided to allow these cells to grow.

We hope this clarifies how we are thinking about this topic and we have added some of these points into the discussion of the manuscript, as well as an overview of this model in Supplementary Fig. 14..

69. It would be helpful to include references to the recent papers on HS regulation of cytokine function and related review (see below). Comparing and/or contrasting the mechanism proposed here with these recent studies on how HS regulated cytokine/chemokine function in other contexts would strengthen the discussion.

Yes we agree! Several immunologists have discovered/are discovering this phenomenon with other cytokines, highlighting the importance of considering the ECM when studying the regulation of (auto)immunity. We will include these citations in our references and discuss them in our text. Thank you for collecting them for us!

<https://www.nature.com/articles/s41590-023-01420-5>

We added this reference about IFN-gamma to our discussion. Please see page 19.

<https://pubmed.ncbi.nlm.nih.gov/36640356/>

We added this reference about CXCL4 to our discussion. Please see page 19.

<https://www.science.org/doi/10.1126/sciimmunol.add1728?s=09>

We added this reference regarding IL-21 binding HS to our discussion. Please see page 19.

<https://www.science.org/doi/10.1126/science.abp8964>

We added a citation on this review to our discussion. Please see page 19.

REVIEWERS' COMMENTS

Reviewer #1 (Remarks to the Author):

The critiques of this reviewer have been well addressed.

Reviewer #2 (Remarks to the Author):

The authors answered all my comments and the manuscript is suitable for publication.

Reviewer #3 (Remarks to the Author):

The authors have fully addressed all of my concerns, congratulations on a fantastic paper that will be very important to the field!